# Simultaneous capturing phonon and electron dynamics in MXenes

Qi Zhang[1], Jiebo Li [2] ✉, Jiao Wen[3], Wei Li[4], Xin Chen [4], Yifan Zhang[2], Jingyong Sun[3], Xin Yan[5], Mingjun Hu[3], Guorong Wu [1], Kaijun Yuan [1,6] ✉, Hongbo Guo[3] ✉ & Xueming Yang [1,6,7]

Plasmonic MXenes are of particular interest, because of their unique electron and phonon structures and multiple surface plasmon effects, which are different from traditional plasmonic materials. However, to date, how electronic energy damp to lattice vibrations (phonons) in MXenes has not been unraveled. Here, we employed ultrafast broadband impulsive vibrational spectroscopy to identify the energy damping channels in MXenes ($Ti_3C_2T_x$ and $Mo_2CT_x$). Distinctive from the well-known damping pathways, our results demonstrate a different energy damping channel, in which the $Ti_3C_2T_x$ plasmonic electron energy transfers to coherent phonons by nonthermal electron mediation after Landau damping, without involving electron-electron scattering. Moreover, electrons are observed to strongly couple with $A_{1g}$ mode (~60 fs, 85–100%) and weakly couple with $E_g$ mode (1–2 ps, 0–15%). Our results provide new insight into the electron-phonon interaction in MXenes, which allows the design of materials enabling efficient manipulation of electron transport and energy conversion.

The knowledge of the interplay between electron and lattice vibration (phonon) is essential for a thorough understanding of energy transport and conversion pathways from electron to phonons in materials[1–3], and it is indispensable for developing rational guiding theories for advanced 2D materials. After decades of development, versatile and valuable applications have been developed by manipulating nonequilibrium interactions between collective free electron oscillations (surface plasmon, SP) and phonons in typical plasmonic metal nanostructures[4–6]. To date, the well-known energy conversion pathway in noble metal nanostructures is that the photon-induced SP decays to generate nonthermal electrons after Landau damping, the nonthermal electrons then evolve into thermal electrons through the electron-electron scattering within hundreds of femtoseconds, and finally, thermal electrons relax within a few picoseconds to vibrational

modes via the electron-phonon coupling (Fig. 1 channel I)[7–9]. Another energy damping pathway has been revealed in the 2D material graphene, where photo-induced plasmons directly convert into intrinsic optical phonons within 20 fs (Fig. 1 channel II)[10]. Different from noble metal nanomaterials and semi-metallic 2D graphene, the 2D plasmonic transition metal carbide materials (MXenes) have broadband electronic absorptions and intricate vibrational modes[11–13], offering more channels for light manipulation. The ultrafast electron diffraction experiment reported the excited lattice vibration within ~230 fs for $Ti_3C_2T_x$, indicating the strong electron-phonon coupling[14]. The time scale observed in their work is close to the electron-electron scattering process (~100 fs)[8], suggesting the partial electrons damp without experiencing the electron-electron scattering. In other words, the unique structures of MXenes may provide competing energy

[1]State Key Laboratory of Molecular Reaction Dynamics and Dalian Coherent Light Source, Dalian Institute of Chemical Physics, Chinese Academy of Sciences, 457 Zhongshan Road, Dalian 116023, P.R. China. [2]Institute of Medical Photonics, Beijing Advanced Innovation Center for Biomedical Engineering, School of Biological Science and Medical Engineering, Beihang University, Beijing 100191, P.R. China. [3]School of Materials Science and Engineering, Beihang University, Beijing 100191, P.R. China. [4]GuSu Laboratory of Materials, Suzhou 215123 Jiangsu, China. [5]School of Mechanical Engineering and Automation, Beihang University, Beijing 100191, P. R. China. [6]Hefei National Laboratory, Hefei 230088, China. [7]Department of Chemistry, College of Science, Southern University of Science and Technology, Shenzhen 518055, P. R. China. ✉e-mail: jiebo39@buaa.edu.cn; kjyuan@dicp.ac.cn; guo.hongbo@buaa.edu.cn

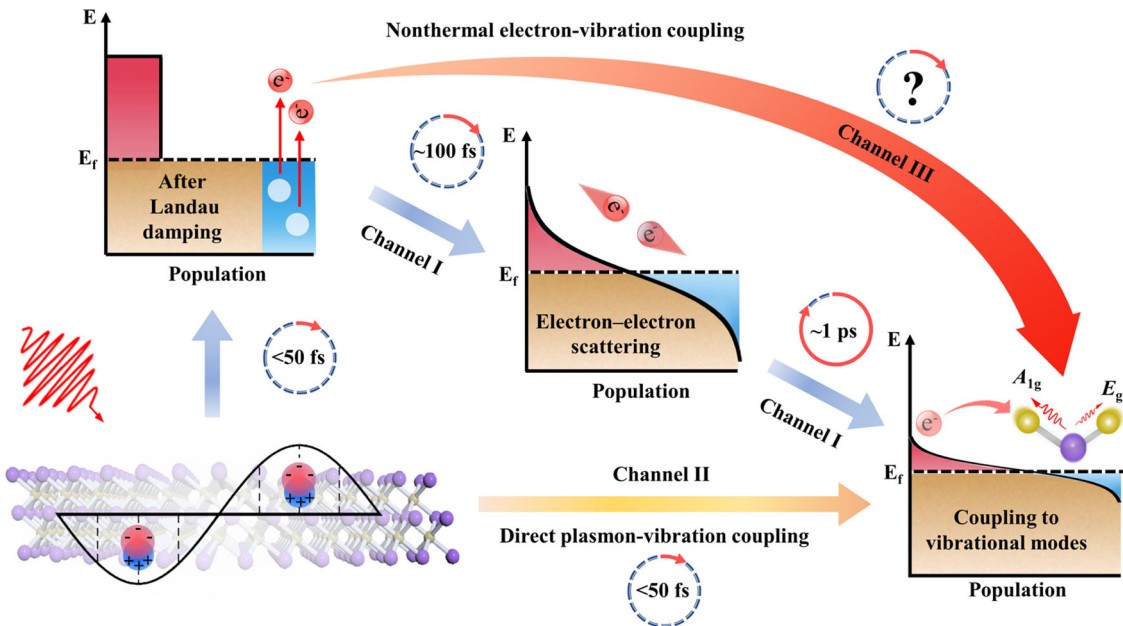

**Fig. 1 | Schematic illustration of three different surface plasmon (SP)-vibration coupling channels after optical excitation.** Channel I: The SP decays to generate nonthermal electrons after Landau damping, the nonthermal electrons then evolve into thermal electrons through the electron-electron scattering, and thermal electrons relax to phonons via the electron-phonon coupling; Channel II: The SP directly converts into phonons; Channel III: The SP decays to generate nonthermal electrons, then the nonthermal electrons directly couple with phonons.

migration pathways from nonthermal electrons (generated interband or intraband transition electrons after Landau damping, but before the electron-electron scattering) directly coupling with specific lattice vibrations (Fig. 1 channel III). However, recent experimental results using the time-resolved terahertz techniques showed the weak electron-phonon coupling in the MXene[15]. These interesting observations reported in the previous works prompt us to reinvestigate both the phonon and electron dynamics and to unravel the different energy transfer channel.

In this work, we use the broadband impulsive vibrational spectroscopy (IVS)[16–18] and the transient absorption spectroscopy to interrogate phonon and electron dynamics simultaneously, which could unravel the entangled pathways from the electron damping to coherent lattice vibration in two typical plasmonic MXenes ($Ti_3C_2T_x$ and $Mo_2CT_x$). Two distinctive coherent vibration responses are observed for the two 2D materials with the SP and IBT (inter-band transient) excitation. For $Ti_3C_2T_x$, the SP and IBT electrons show similar vibrational responses. Furthermore, by controlling the optical parameters of the pump pulses (wavelength and fluence), our study reveals a different energy transfer pathway (Fig. 1 channel III) from the SP damping to the nonthermal electrons, followed by the nonthermal electrons directly strong coupling with the $A_{1g}$ mode within ~60 fs and weak coupling with the $E_g$ mode during 1–2 ps under the SP excitation. Meanwhile, the excited IBT nonthermal electron also could directly couple with the $A_{1g}$ mode without the electron-electron scattering. In contrast, the resonant and nonresonant SP excitations in $Mo_2CT_x$ show different vibrational dynamical responses, which suggests a direct SP-phonon coupling under the SP excitation. The energy pathway is determined by electron-phonon coupling time and electronic density of states near the Fermi surface.

## Results

### Phonon and electron dynamics in ultrafast spectroscopy

As illustrated in Fig. 2a, once a certain ultrafast photon is injected into a material by the pump light, it leads to deformation of the electronic cloud in the material. This photoinduced potential energy change drives atomic displacement[19–22]. In other words, light-induced electronic excitation in 2D materials could couple with internal vibrational modes (phonons). This interplay thus appears in the transient spectra as the time-dependent oscillation. In this work, we sought to reveal the damping pathways of the SPs in two MXenes, $Ti_3C_2T_x$ and $Mo_2CT_x$ (Characterizations are given in Supplementary Figs. 1 and 2). The absorption spectrum of $Ti_3C_2T_x$ (red line in Fig. 2b) shows an SP resonance at ~780 nm, while $Mo_2CT_x$ (green line Fig. 2b) presents an SP absorption state at ~532 nm. If the energy of the excited electrons flows to a certain lattice vibration mode, coherent oscillations of the lattice vibration should be observed. Therefore, the dynamic process of the electron relaxation would display a combination of decay and oscillation, as shown in Fig. 2c. Interestingly, due to the different electronic states generated by the pumping pulse, the distinctly coherent coupling between these electronic states and the lattice vibrations can be observed via the broadband detection, as shown in the inset of Fig. 2c. The electronic and vibrational damped signals (Fig. 2d) can be numerically separated by the multiexponential and oscillation fitting[16,23,24]. The frequencies of coherent lattice vibration modes can be obtained by fast Fourier transform (FFT) of the oscillation curve (Fig. 2e). Therefore, we could determine a scenario for the energy flow from a stimulated SP to the specific atomic motions by exploring the interplay between electron and coherent phonon (CP) vibration dynamics with ultrashort laser pumping. Compared with the previous studies of MXenes[13–15,25,26], the IVS could capture phonon and electron dynamics information simultaneously.

### The observations of phonon and electron dynamics

To determine the time-resolved coherent vibrational distribution of $Ti_3C_2T_x$ with different electronic excitation states, we selected to pump the SP state at 780 nm and the IBT state at 532 nm. Detailed analyses of the probe-dependent electron/phonon dynamics and phase shifts are presented in Supplementary Figs. 3–5 and Supplementary Note 1. Apparently, one CP state was detected in the wavelength regions (i) and (ii) shown in Fig. 2c, while two CP states existed in the wavelength region (iii). The electron dynamics (Supplementary Fig. 6) and the phonon response (Fig. 3a, b) could thus be revealed with the two pump pulses. The formation and decay time constants of electrons and

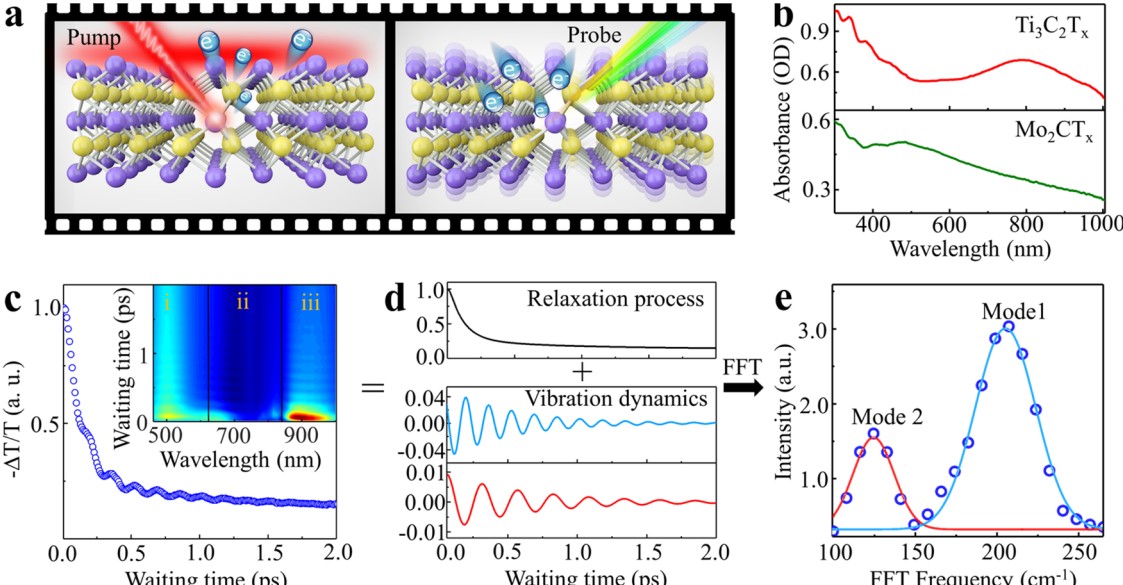

**Fig. 2 | Concept of the broadband impulsive vibrational spectroscopy (IVS) experiment with plasmonic MXenes. a** The scheme of the pump-probe IVS used to measure the SP-vibrational coupling and vibrations of $Ti_3C_2T_x$. The pump pulse generates excited electrons, and then the dynamics of electrons and lattices were monitored by the probe pulse. **b** Absorption spectra of plasmonic MXenes: $Ti_3C_2T_x$ (red line) and $Mo_2CT_x$ (green line) films. **c** The pump-probe data detected at 900 nm show photoinduced differential transmission as a function of the time delay. The inset shows a 2D map of the photoinduced differential transmission. **d** Fitting results from a differential transmission signal consisting of relaxation and oscillating components, which are equivalent to the vibration mode frequency. **e** The vibrational spectrum was obtained from the fast Fourier transform (FFT) of differential transmission data after subtraction of the relaxation process.

phonons are obtained by fitting with formulas (1) & (2) (See Materials and Methods data analysis). Noticeably, the CP dynamics in Fig. 3a, b (with the IBT and SP excitation, respectively) show similar formation ($67 \pm 20/55 \pm 20$ fs) and decay ($820 \pm 60/680 \pm 42$ fs) time constants. The similarity suggests the SP decay to generate IBT electrons, followed by the electron coupling to CP, rather than the SP direct coupling to CP. This is reasonable because the screening of nucleus by free electrons usually prevents the direct plasmon-phonon couplings[27]. The $Ti_3C_2T_x$ with metallic electronic structure has a lot of free electrons near the Fermi surface. Thus, the SP mainly interacts with the free electrons near the Fermi surface, which effectively screens the direct interaction between the SP and the phonon.

To identify whether the electron-phonon coupling occurs after the electron-electron scattering, we determined the electron-electron scattering half-time to be ~50 fs (the details of acquiring the half-time are described in Supplementary Figs. 7 and 8 and Supplementary Note 2), and performed the pump fluence-dependent experiments to obtain the electron and CP dynamics. Figure 3c, d display that the electron relaxation slows down gradually with increasing the pump fluence. Such phenomenon is similar to that observed in typical noble metals, which has been attributed to the electron temperature elevation (*i.e.*, electron thermalization) induced by the electron-electron scattering[28–30]. However, the formation time and the phase of CP (Fig. 3c, d and Supplementary Figs. 9 and 10; marked lines showing no obvious phase shift in CP) are independent with the pump fluence, indicating that the CP stems from the nonthermal electrons rather than the thermal electrons. This interpretation is confirmed by a control experiment employing the sample of gold nanorods (GNRs), in which the dynamic process is unambiguously assigned to the channel I (Fig. 1). The oscillating dynamic traces of GNRs (see Supplementary Fig. 11) exhibit the obvious phase shift and the CP dynamics variation with increasing the pump fluence (see Supplementary Note 3). On the other hand, the linear fit of the pump fluence-dependent electron-phonon coupling time constants gives the intercepts[29,31], as shown in the inset panels of Fig. 2c, d. The intercept values ($57 \pm 10$ and $39 \pm 10$ fs under pumping at 532 and 780 nm respectively) represent the

electron-phonon coupling time constant without the thermal electron generation process in $Ti_3C_2T_x$. These time constants are consistent with the above-mentioned formation time constants of CP ($67 \pm 20$ and $55 \pm 20$ fs), which further verifies that the CP originates from the nonthermal electrons. It is also noted that the electron-phonon coupling time constants are close to the electron-electron scattering time constants (~50 fs), suggesting that the two physical processes occur simultaneously. Based on the time constants, almost half of the nonthermal electrons directly couple with the CP. In summary, the experimental observations demonstrate a distinct energy transfer pathway (Fig. 1 channel III), i.e., the SP damps to nonthermal electrons, and a significant portion of them directly couple with the CP without experiencing the electron-electron scattering.

The time-resolved vibrational spectra of $Ti_3C_2T_x$ show only one mode at 200 cm⁻¹ by pumping at 532 nm (Fig. 3e). However, the plasmonic excitation at 780 nm generates a strong vibrational mode at ~200 cm⁻¹ and a weak one at ~128 cm⁻¹ (Fig. 3f). The FFT results of each 0.5 ps interval (Supplementary Fig. 12) show that the mode at ~128 cm⁻¹ appears during 1.0–1.5 ps. The fitting result in Fig. 3a shows that the coherent oscillation exclusively comes from the ~200 cm⁻¹ vibrational mode for the pump pulse at 532 nm. In contrast, for the pump excitation at 780 nm, the ~128 cm⁻¹ vibrational mode contributes to ~15% population of the coherent oscillation, and the ~200 cm⁻¹ vibrational mode still has a major contribution. These two vibrational modes are consistent with the Raman spectra shown in Fig. 3g, h, which are assigned as an out-of-plane ($A_{1g}$, ~200 cm⁻¹) and an in-plane ($E_g$, ~128 cm⁻¹) vibrations of titanium atoms in the outer layer (Ti1/Ti2) as well as those of carbon and surface groups[32–34]. There is no observable frequency shift of the vibrational modes in Raman and IVS experiments, which may be due to the limitation of the time resolution (~40 fs) in our experimental system. It is noted that the signal of the $E_g$ mode in IVS (Fig. 3f and Supplementary Fig. 12) is very weak, differing from the strong signal of $E_g$ in the Raman spectrum (Fig. 3h). This is probably because the Raman spectra can only detect zone-center optical phonons[35]. The best fit to the IVS experimental data demonstrates that the electrons

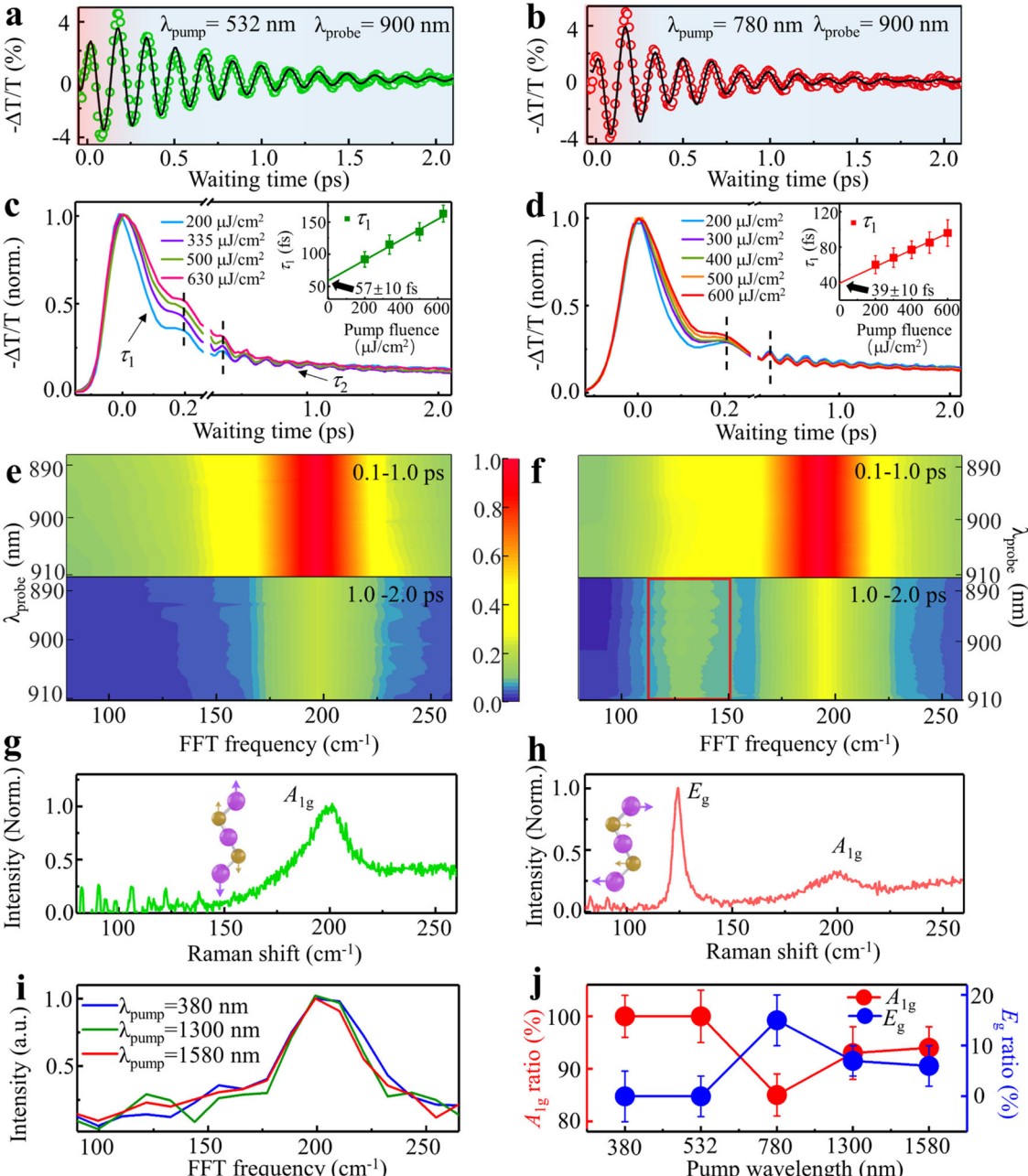

**Fig. 3 | The coherent phonon (CP) dynamics, time-resolved IVS, and Raman spectra of a Ti$_3$C$_2$T$_x$ film with different optical excitations. a, b** CP signals (green and red circles) were monitored at 900 nm after excitation at 532 and 780 nm, respectively. The two black lines show fits to the CP data. **c, d** The 532 and 780 nm pump fluence dependent dynamics data were monitored at 900 nm. **e, f** Time-resolved FFT maps for 0.1–1.0 and 1.0–2.0 ps with different excitation wavelengths. **g, h** Raman spectra of the Ti$_3$C$_2$T$_x$ film under the 785 nm and 532 nm pump. **i** FFT vibration spectra obtained from the coherent dynamics data in Supplementary Fig. 14 over 1.0–2.0 ps with pumping at 380, 1300 and 1580 nm. **j** Branching ratios of the two modes with various near-infrared pumps were extracted from the data of Fig. 3a, b and Supplementary Fig. 14. And relative ratios of the $A_{1g}$ and $E_g$ modes respectively originating from $A_{1g}/(A_{1g} + E_g)$ and $E_g/(A_{1g} + E_g)$ were acquired by using formula (2) to fit. The error bars represent standard deviation.

strongly couple with the $A_{1g}$ mode (~60 fs, 85–100%) and weakly couple with the $E_g$ mode (1–2 ps, 0–15%).

To further confirm the conclusions, we performed pump wavelength-dependent experiments. The near-infrared absorption spectrum of Ti$_3$C$_2$T$_x$ has been displayed in Supplementary Fig. 13. The summary results with excitations at UV (380 nm) and near-infrared (1300 and 1580 nm) wavelength regions are shown in Fig. 3i, j and Supplementary Fig. 14. The $E_g$ vibrational mode is not active after excitation at 380 nm, although the band at 380 nm has the same absorbance as that at 780 nm in Ti$_3$C$_2$T$_x$ (Fig. 2b). Then, we tuned the pump wavelength to match the longitudinal SP bands (1300 and

1580 nm)[11]. The results (Fig. 3i) show a weak $E_g$ vibration peak at 128 cm$^{-1}$. The relative population (<8%) of the excited $E_g$ mode is still lower than the population (~15%) observed with pumping at 780 nm. The vibration response of Ti$_3$C$_2$T$_x$ as a function of the pumping wavelength is summarized in Fig. 3j, which clearly demonstrates that the electron strongly couples with the $A_{1g}$ mode and fairly weakly couples with the $E_g$ mode.

In contrast, the electron and phonon dynamics in Mo$_2$CT$_x$ (Supplementary Fig. 15 and Note 4) show that the time-resolved oscillatory signals (Supplementary Fig. 15a) are different following excitation at 532 and 700 nm. The intensity of the CP reaches its maximum in the

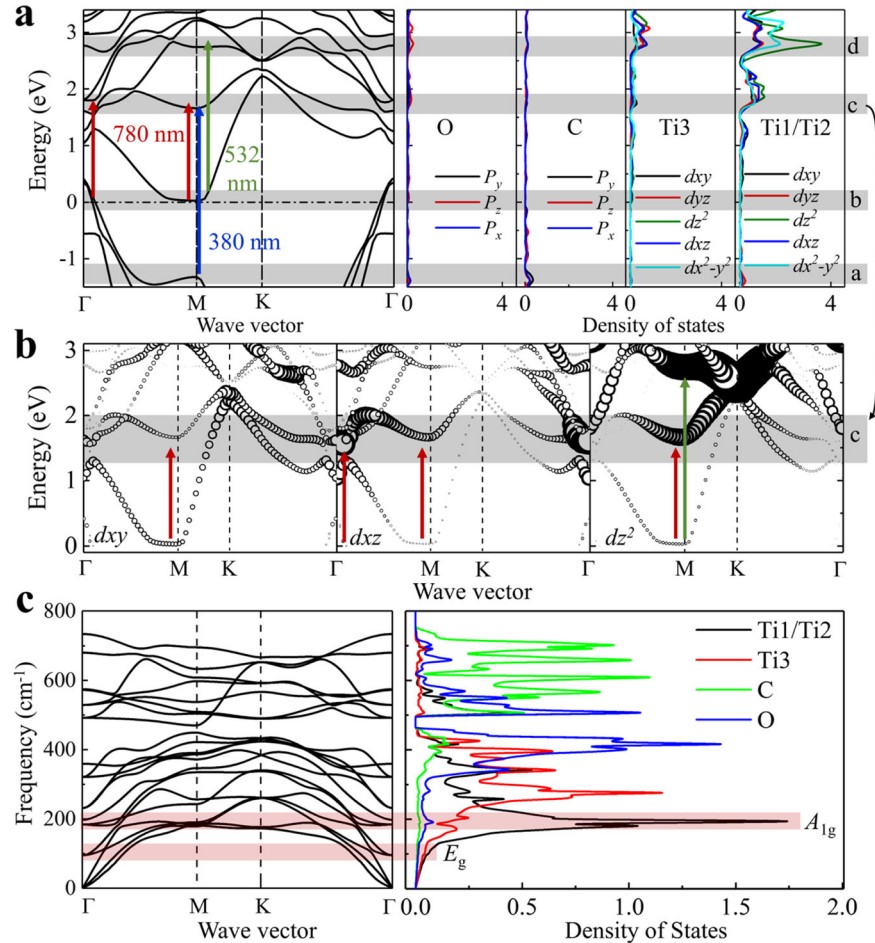

**Fig. 4 | The calculated band structures and density of states (DOS) of Ti₃C₂Tₓ (Ti₃C₂O₂). a** Hybrid functional electron structures and DOS. Marked around −1.3 eV as the a band, around −0.3 eV as the b band, and about 1.6 eV as the c band (main contribution of Ti1/Ti2 $dz^2$, $dxz$, $dxy$ orbitals), -2.6 eV as the d band (main contribution of Ti1/Ti2 $dz^2$ orbital). **b** Hybrid functional fatband of $dxy$, $dxz$ and $dz^2$ of Ti (The diameters of the circles represent the intensity of DOS). The different colored arrows indicate possible electronic transitions after excitation with different pump-wavelengths. **c** The phonon dispersions band and DOS. The mark with bar indicated higher DOS of $A_{1g}$ than $E_g$.

first oscillation period for the SP resonant excitation (532 nm). Then, the CP loses its approximately half population within 230 fs, and the remaining population decays slowly within 2 ps. For the nonresonant excitation (700 nm), the CP shows a formation time of 163 ± 30 fs and a relaxation time of 870 ± 50 fs. The time-partitioned FFT spectra of Mo₂CTₓ (Supplementary Fig. 15b) display two modes (-150 and ~300 cm⁻¹) under the resonant excitation and one mode (~300 cm⁻¹) under the nonresonant excitation. This reveals the selective interaction between electrons and phonons in MXenes.

### Orbital-dependent electron structure and phonon dispersions band

To gain insight into the pump wavelength-dependent electron-phonon coupling, we calculated the electron structure and density of state (DOS) of Ti₃C₂O₂ (Fig. 4a). We marked with a gray bar around −1.3 eV (0.0 eV, 1.6 eV, 2.6 eV) and named a (b, c, d) band. The c band (around 1.6 eV) is contributed mainly by the $dxy$, $dxz$, and $dz^2$ orbitals of Ti1/Ti2 contribute. The d band is contributed mainly by the $dz^2$ orbitals of Ti1/Ti2. Figure 4b shows fat bands of $dxy$, $dxz$, and $dz^2$ of Ti (The diameters of the circles represent the intensity of DOS). Figure 4c shows the phonon dispersions band and DOS. Comparing calculations (Fig. 4a) and experiments after excitation at 380 nm ($\hbar\omega_{pump}$ = 3.3 eV) and 532 nm ($\hbar\omega_{pump}$ = 2.3 eV) shows that the a-c and the b-d transition occur at the M point and electrons can be excited to Ti $dz^2$ orbital. In contrast, for the pump at 780 nm ($\hbar\omega_{pump}$ = 1.6 eV), the b-c transition

occurs at M and Γ points, and electrons can be excited to $dxy$, $dxz$, and $dz^2$ orbital. According to orbital-dependent fatband electron structure (Fig. 4b), the DOS is relatively lower in $dxy$ orbital in the c band at M and Γ point than in $dz^2$ and $dxz$ orbitals, suggesting the transitions of electrons from the ground state into $dz^2$ and $dxz$ orbitals are two main routes after excitation at 780 nm. Besides, the calculated results (Fig. 4c) show that the DOS of the $A_{1g}$ mode at -200 cm⁻¹ is much higher than the DOS of the $E_g$ mode at -128 cm⁻¹.

## Discussion

Our results revealed the characteristics of the photon energy conversion pathway for the plasmonic MXenes by tuning the pump laser wavelength and fluence, and by changing the metal elements of materials. In the following, we discuss the symmetry-dependent electron-phonon coupling and the damping SP channels.

Firstly, the electron-phonon interactions are dependent on the spatial symmetry of atom displacement and electronic cloud deformation by the optical field. The symmetry match between electronic orbitals and vibrational mode is crucial for electron-phonon coupling. As shown in Fig. 5a, the movement along the z-direction of $dz^2$ orbital could modulate the electronic cloud to strengthen the coupling of the out-of-plane vibration mode (such as $A_{1g}$ mode) but has no effect on the coupling of the in-plane vibrational mode (such as $E_g$ mode). While the movements along the x- and z-direction of $d_{xz}$ orbital can couple with the in-plane and the out-of-plane vibrational modes. For the

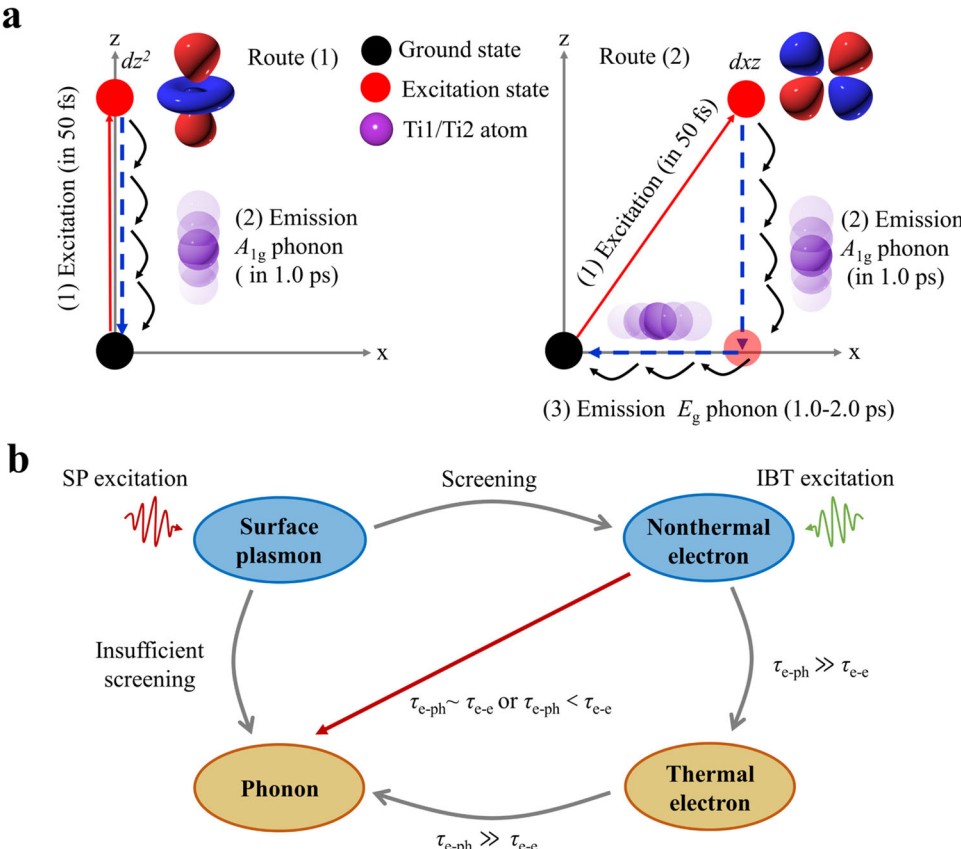

**Fig. 5 | Energy transfer pathways. a** The symmetry-dependent electron-phonon coupling: for route 1, the electron is excited into $dz^2$ orbit, then it couples with $A_{1g}$ mode; for route 2, the electron is excited into $dxz$ orbit, then it couples with $A_{1g}$ and $E_g$ modes. The Shadings of purple ball indicate the directions of atomic vibration. **b** The energy transfer channels from photon to phonon. The $\tau_{e\text{-}ph}$ is the time constant of the electron-phonon coupling and the $\tau_{e\text{-}e}$ is the time constant of the electron-electron scattering. For a SP material after absorbing photons to excite the SP, in a case of a strong screening effect of the nucleus by high free electrons density near the Fermi surface, the SP decays generate the nonthermal electron. These generated nonthermal electron could interact directly with $A_{1g}$ mode if the $\tau_{e\text{-}ph}$ is close to or less than the $\tau_{e\text{-}e}$, in which the energy of the thermal electron will dissipate to acoustic phonons. If the $\tau_{e\text{-}ph}$ is much greater than the $\tau_{e\text{-}e}$, the energy of nonthermal electron would damp to all kinds of phonons mediated by the thermal electron. In case of a weak screening effect (insufficient screening), the SP decays could couple directly with phonon.

Ti$_3$C$_2$T$_x$ under the 780 nm pump, electrons in two orbitals ($dxz$ and $dz^2$) are allowed to couple with the $A_{1g}$ mode, while electrons in $dxz$ are only allowed to couple with the $E_g$ mode. Under the 532 or 380 nm pump, electrons are excited mainly into the $dz^2$ orbital, leading to strongly coupling with $A_{1g}$ and barely coupling with $E_g$. Furthermore, a high DOS of a phonon mode is a key factor for the electron-phonon coupling, as it ensures sufficient quantities of phonons interplay with electrons. Figure 4c shows that the DOS of the $A_{1g}$ mode at ~200 cm$^{-1}$ is much higher than the DOS of the $E_g$ mode at ~128 cm$^{-1}$, indicating the electron prefers to couple with the $A_{1g}$ mode. The parallel dynamics studies of Mo$_2$CT$_x$ and calculation (Supplementary Figs. 15–16, and Note 4) show that the electron-phonon coupling also obeys the symmetry rule. Furthermore, the fast CP response (Supplementary Fig. 15a) in Mo$_2$CT$_x$ suggests a direct SP-CP interaction under the SP excitation. Compared with the metallic Ti$_3$C$_2$T$_x$, the electron density near the Fermi surface is low in semi-metallic Mo$_2$CT$_x$[12,36]. Thus, free electrons are not sufficient to screen the direct SP-CP interaction. It is obvious that the electron density near the Fermi surface is a crucial factor in modulating the relaxation channel of SP.

Secondly, the electron-phonon coupling is related to the electron density of states near the Fermi surface and the matching between electron and phonon energy levels[37,38]. Metallic MXenes have higher densities of electronic states near the Fermi surface. The plasmon band (on the order of 10$^4$ cm$^{-1}$) is three or four orders of magnitude larger than the phonon modes (on the order of 10$^0$–10$^1$ cm$^{-1}$)[7,8] in typical metal plasmonic materials. This large energy gap requires that the plasmonic energy engages in several sequential physical processes, such as Landau damping, electron-electron scattering, and electron-phonon coupling. In contrast, the plasmonic band and the high-frequency phonons in graphene are both located in the mid-infrared region (on the order of 10$^4$ cm$^{-1}$)[10,39]. Unlike Au nanostructures and graphene, MXenes (Ti$_3$C$_2$T$_x$ and Mo$_2$CT$_x$) present plasmonic modes with frequencies ranging from the visible to mid-infrared (10$^3$–10$^4$ cm$^{-1}$)[8] regions and high phonon frequencies (10$^2$–10$^3$ cm$^{-1}$)[11,32,34]. Thus, the plasmons band frequencies are larger than the phonons by one or two orders of magnitude. The gap of frequency between plasmon and phonon is significantly smaller in MXenes than in metal nanostructures and is slightly larger than in graphene. Better energy level matching also should enhance electron-phonon coupling.

Thirdly, electron-phonon coupling was affected by material components. MXenes have various metal elements as well as ionic and polar structures. Ti and Mo metal elements mainly contribute to the presence of electronic states near the Fermi level for the two MXenes. The electron-phonon interactions time constants in metals are inversely proportional to atomic mass (metal atomic masses: Ti<Mo<Au)[40,41]. Our results show that the electron-CP coupling strengths of Ti$_3$C$_2$T$_x$ (electron-phonon coupling time constant: ~50 fs) and Mo$_2$CT$_x$ (electron-phonon coupling time constant: ~152 fs, Supplementary Fig. 15c) are stronger than that of the plasmonic Au nanostructure (electron-phonon coupling time constant: 1–2 ps)[28,42].

These fast coupling time constants in MXenes suggest strong electron-phonon interactions. Considering the ionic and the polar structure of MXenes, the electron-phonon interactions generally originate from the deformation potential and the Fröhlich mechanism[43]. The two electron-phonon coupling mechanisms usually dominated in polar semiconductive 2D materials such as halide perovskites[2,24,44], single-$MoSe_2$[45] and typical $MoS_2$[46,47]. Transverse optical phonon mode (TO) mainly infers to the short-range deformation potential induced electron-phonon coupling[48]. The longitudinal optical phonon mode (LO) can generate long-range macroscopic electric fields, which leads to long-range Fröhlich interaction induced electron-phonon coupling[49]. The previous study using terahertz spectroscopy[15] suggested the weak long-range electron-LO coupling whereas the ultrafast electron diffraction experiment[14] purposed the strong electron-phonon coupling without clarifying the energy transfer pathway and the coupled phonon modes, which has been clearly presented in this work. In metals, only short-range electron-phonon interactions are effective since the free electrons in the conduction band could screen the long-range electric potential of the displaced nuclei[50], indicating that the strong electron-TO coupling is allowed but the electron-LO coupling is weak in $Ti_3C_2T_x$. Furthermore, we determined the electron-phonon coupling constant ($\lambda$) in the framework of metals[40]. The values of $\lambda$ are $1.62 \pm 0.33$ for $A_{1g}$ and $0.054 \pm 0.018$ for $E_g$ in $Ti_3C_2T_x$ from the measured electron and CP dynamics. The details of obtaining $\lambda$ can be found in the Supplementary Note 5. Compared with the values ($\lambda$ ~ 0.03–1.45) of most metals and semi-metals, such as Au, Cu, Ti, V, Nb, W, Pb[41] and Ag, Al, TiAl[37] and graphene[51], the electron-phonon coupling constant in $Ti_3C_2T_x$ indicates a strong electron-$A_{1g}$ phonon coupling and a weak electron-$E_g$ phonon coupling. Additionally, the significant frequency-shift of the $A_{1g}$ mode has not been observed in $Ti_3C_2T_x$ in our experiment, which is similar to the single-layer $MoSe_2$ of strong electron-$A'_1$ phonon coupling[45] and the $NiPS_3$ of the strong electron-$A_{1g}$ phonon coupling[52]. Therefore, the metallic electron structure and the polar lattice mode in MXenes play key roles in electron-phonon interactions. Based on the discussions, the general energy migration pathways are epitomized in Fig. 5b.

In summary, simultaneous capturing phonon and electron dynamics allows us to reveal the pathways from photon energy to phonon in MXenes. A distinct energy transfer pathway (The SP decay generates nonthermal electrons, followed by nonthermal directly strong coupling to the $A_{1g}$ mode in $Ti_3C_2T_x$) is identified, which is distinct from the typical damping pathways in the traditional SP materials. Furthermore, the energy pathways about excited electrons can be modulated by electron-phonon coupling and electronic density of states near the Fermi surface. The mechanism of the energy transfer and the symmetry-dependent electron-phonon coupling allows scientists to utilize photon energy efficiently, and it is also vital for developing new insights into different plasmon physics and subsequent developments and industrial applications related to new photothermal therapy, chemical reaction, and optoelectronic devices.

## Methods

### Synthesis of $Ti_3C_2T_x$
The synthesis of $Ti_3C_2T_x$ (Shandong Xiyan New Material Technology Co., Ltd.) is as follows[53–55]: In detail, 1.6 g of LiF was added slowly to 20 mL of 9 M hydrochloric acid, and the solution was stirred in a polytetrafluoroethylene flask for 5 min. Then, 1 g of $Ti_3AlC_2$ powder was added slowly to the etching solvent over 10 min and then stirred at 35 °C for 24 h. After the reaction was complete, the product was washed with 1 M hydrochloric acid and centrifuged to remove excess LiF. Then, it was washed with deionized water several times and subsequently centrifuged (3500 rpm for 5 min) 5 – 7 times until the pH of the solution reached 7. Afterward, the centrifugal solution was dispersed into deionized water, sonicated for 3 h under a flow of argon gas, and centrifuged at 3500 rpm for 1 h to obtain the single- or few-

layer $Ti_3C_2T_x$ solution. The GNRs in the aqueous solution as a control sample were purchased from Jiangsu XFNANO Materials (Nanjing, China).

### Synthesis of $Mo_2CT_x$
$Mo_2CT_x$ (Shandong Xiyan New Material Technology Co., Ltd.) was synthesized by etching $Mo_2Ga_2C$ with a LiF–HCl mixed solution[56]. LiF (1.8 g) was added slowly to 20 mL of 12 M hydrochloric acid. Then, 1 g of $Mo_2Ga_2C$ was slowly immersed into the etching solvent with a Teflon-coated magnet for 168 h at room temperature. After the reaction was complete, the product was washed with 1 M hydrochloric acid and centrifuged to remove excess LiF. Then, the mixture was centrifuged and washed with deionized water several times until the pH reached 7. The obtained powder was freeze-dried in a vacuum overnight. Finally, the sediments were redispersed into deionized water, sonicated for 30 min under a flow of argon gas, and centrifuged at 3500 rpm for 1 h to obtain the $Mo_2CT_x$ solution.

### Preparation of $Ti_3C_2T_x$ and $Mo_2C_2T_x$ films
Samples of $Ti_3C_2T_x$ and $Mo_2C_2T_x$ films were obtained by dripping -0.2 mg mL$^{-1}$ $Ti_3C_2T_x$ and $Mo_2C_2T_x$ aqueous solutions onto a $CaF_2$ window and vacuum drying after 6 h. To avoid light-induced oxidation of the two samples, they were placed into a vacuum cell (Oxford instruments, Optistat DN-V) during measurement of the broadband IVS and pump-probe data.

### Broadband IVS and transient absorption spectroscopy and data analysis
These experiments were conducted on a Ti: sapphire laser system (coherent Vitesse and Coherent Legend Elite He+USP-1K-III) with an output of 7.1 mJ for 35 fs pulses at 800 nm and a 1 kHz repetition rate. The pump beam was generated by an optical parametric amplifier that radiated tunable optical pulses within the range 240–2400 nm. The broadband white light continuum used as the probe pulse was produced by focusing a weak energy part of the source pulse on a sapphire plate. The time delay between pump and probe pulses was achieved with a computer controlling a motorized delay line in the pump pulse optical path. Clear oscillation signals can be achieved once the step length is set up to 10–20 fs. To record the time-resolved transmission signal, the pump pulse was blocked alternately by a synchronized chopper. The IRF (instrument response function) of this measurement was determined to be 40–50 fs by measuring substrate ($CaF_2$) as shown in Supplementary Fig. 17. Electronic relaxation signals were deconvoluted by fitting with the formula:

$$S_e(t) = IRF \bigotimes \left( A^* e^{-\frac{t}{\tau_1}} + B^* e^{-\frac{t}{\tau_2}} \right) \quad (1)$$

where $S_e(t)$ is the electron signal intensity with time delay. A and B and $\tau_1$ and $\tau_2$ are amplitudes and time constants of the two exponential decay components.

The oscillating components signals are well fitted with:

$$S_c(t) = \sum_{i=1,2} A_i \cos(\omega_i t + \theta_i)^* \left( -e^{-\frac{t}{\tau_{fi}}} + e^{-\frac{t}{\tau_{ri}}} \right) \quad (2)$$

where $S_c(t)$ is the oscillating components signals with time delay. $A_i$ is the oscillation amplitude, $\omega_i$ is the characteristic frequency of phonon. $\theta_i$ is the phase shift. $\tau_{fi}$ is the formation/response time constant for appearance excited CP and $\tau_{ri}$ is the coherent phonon relaxation constant. More details of data analysis and correcting for the chirp of white light and error analysis has been shown in Supplementary Notes 6, 7 and Supplementary Figs. 18–20. Moreover, the global analysis[57] of kinetic were shown in Supplementary Fig. 21 and fitting multi-wavelengths analysis oscillatory data were shown in Supplementary Fig. 22. And the

polarization dependent dynamics of $Ti_3C_2T_x$ and detail data of $Mo_2CT_x$ are shown in Supplementary Figs. 23–28.

## DFT computational

The DFT calculations were performed using the projector-augmented wave method and the Perdew-Burke-Ernzerhof exchange-correlation functional as implemented in the Vienna Ab initio Simulation Package[58,59]. In addition, the hybrid functional calculations based on the Heyd-Scuseria-Ernzerhof (HSE06) exchange are further used for comparison of reliable band structure. The HSE06 functional is based on a screened Coulomb potential applied to the exchange interaction with the screening parameter of 0.2[60]. The plane-wave energy cutoff was set to be 550 eV, and the 2D Brillouin zone was sampled using a $38 \times 38 \times 1$ mesh per unit cell. In order to avoid interactions between the periodic images (the single layer and its images along the z-direction), the calculations were performed with a large unit cell including ~20 Å thick vacuum space. A semiclassical dispersion correction scheme, DFT-D3, was employed to describe van der Waals interactions[61]. All the atoms were allowed to relax until the forces exerted on each atom were $<10^{-5}$ eV/Å during structural optimization. The phonon dispersion plots were obtained using the force constants method as implemented in the PHONOPY package[62]. Force constants are prepared for building the dynamical matrices with $7 \times 7 \times 1$ Mon-khorstPack k mesh in a $5 \times 5 \times 1$ supercell. $Ti_3C_2O_2$ and $Mo_2CT_x$ were chosen to perform the calculation because most of the synthesized MXene end with O- terminal groups[63]. And the five d-ortials of $Ti_3C_2O_2$ are displayed in Supplementary Figs. 29.

## Data availability

The authors declare that the data supporting the findings of this study are available within in this published article and its supplementary. Source data are provided with this paper.

## Code availability

The codes used to process and analyze the data presented in this work are available as Supplementary Software.

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

## Acknowledgements

The experimental work was supported by the National Natural Science Foundation of China (Grant Nos. 22073003, 21922306, 21873099, 21803006), the Chemical Dynamics Research Center (Grant No. 22288201), the Key Technology Team of the Chinese Academy of Science (Grant No. GJJSTD20220001), the Innovation Program for Quantum Science and Technology (2021ZD0303304), the Liaoning Revitalization Talents Program (Grant No. XLYC1907154), and the Fundamental Research Funds for the Central Universities.

## Author contributions

Q.Z., J.L., K.Y. and H.G. designed the experiments. Q.Z. performed ultrafast experiments. J.W. written the code. Q.Z, J.W., Y.Z. J.L. and K.Y. analyzed the ultrafast experiments data. W.L. and X.C. contributed the theoretical calculations. Y.Z. contributed the numerical analytical Fig. 3e, f and M.H. analyzed the characterization data of MXenes. J.S, X.Y., G.W and X.Y. discussed the results. All authors participated in writing of the manuscript.

## Competing interests

The authors declare no competing interests.
