## [Peer Review File · Nature Communications]

Simultaneous capturing phonon and electron dynamics in MXenesREVIEWER COMMENTS

Reviewer #1 (Remarks to the Author):

Comments:

Electron-phonon interaction is essential for understanding the hot carrier cooling mechanism in solid state materials. Plasmonic MXenes aroused researchers' interest recently due to the unique electronic and phononic properties. In this work, the authors attempted to unveil the coupling between electron and phonon in MXenes by tracking the energy flow from charges to phonons, employing ultrafast impulsive vibrational spectroscopy. They found the energy of photoinduced electrons is transferred to coherent phonons without/before involving electron-electron scattering, where the electrons are strongly coupled with the A_{1g} mode (~60 fs) while weakly coupled with the E_g mode (1-2 ps).

I considered this work is systematic and provides a novel aspect to advance our current understanding about the electron-phonon interaction in MXenes. However, the current version of the manuscript has several unclear points or unsupported claims. The authors should clarify them, before the work can be considered for publication in Nature Communications.

1) The authors proposed the seemingly debate whether strong or weak electron-phonon coupling is in MXenes, based on the previous works. [ACS Nano 15, 14071-14079 (2021); Nat. Phys. 1-7 (2022).] However, in the case of metallic Ti₃C₂T_x, M. Volkov et. al. investigated the photoexcited charge carrier (with an excess energy, i.e., hot carriers) interacting with the phonons (in a non-equilibrium situation), while W. Zheng et. al. studied the charge carrier (i.e., without photoexcitation, i.e., cold carriers) scattering with the phonons (in the equilibrium condition). Note that the interaction process is basically different between phonons and (hot or cold) charge carriers. Therefore, I doubt if it makes sense to propose such a "debate".

On the same note: The authors did not come back to this "debate" in the end. Are their results in line with one of the two papers discussed?

2) It is an interesting observation that some specific modes are coupled to the photogenerated charge carriers in the early time scale. While this does provide a good indication for e-ph coupling, it does not provide information on the energy branching into electron-electron scattering and population of coherent phonons by non-thermalized hot carriers. From this perspective, these two energy relaxations are competing in the sub-100 fs: likely both are active and plausible. I don't follow how the author can reach a conclusion to exclude e-e scattering by stating that "our results demonstrate a new energy damping channel, in which the Ti₃C₂T_x plasmonic electron energy transfers to coherent phonons (CPs) by nonthermal electron after Landau damping, without involving electron-electron scattering."

The author aimed to support their claim by fluence dependent studies in a rather indirect way. They claimed that "the formation time and phase of CP (Fig. 3c&d and Supplementary Fig. 7 and 8; marked lines showing no obvious phase shift in CP) are independent of the pump fluence, indicating that the CP stems from nonthermal electrons rather than thermal electrons." I don't follow the argument. In the "linear" excitation regime, the observables or lifetimes can be independent of the fluence. So please clarify it, and this is a very strong claim which requires further support: ideally a control study in a system where thermalization is known to be the main pathway for energy relaxation and compare the data to the current study.

3) In line 181, the authors stated that the electrons are strongly coupled with the specific phonon modes. By comparing the peaks of the phonon modes from Raman spectrum (Figure 3 h) and IVS (Supplementary Fig. 9), one would expect the frequency-shift of the modes due to the strong electron-phonon coupling. However, the spectra seem to overlap well. This applies for time-independent IVS spectra (Supplementary Fig. 9). The discussion should be included in the revised ms.

4) In Figure 2d, what is the nature of the electron relaxation process? The lifetime in this work is much shorter than that reported in the previous work. [J. Phys. Chem. C 2020, 124, 11, 6441–6447] Can the authors provide any explanations?

5) It is difficult to imagine how the E_g and A_{1g} ratio are obtained (as presented in Figure 3j). How is the population (in %) estimated? In figure 3i, with the signal to noise level I don't see any difference between the three data shown. Please clarify

6) The authors should provide the meaning of error bars and how they were obtained, in particular when they compare the nonthermal electron decay time constants and coherence phonon formation time in lines 158-161.

7) The schematics in figure 1 do not reflect the channel III that the author proposed. What it looks like is just an escape from thermalization. It does not show the coupling from nonthermal hot carriers to CP. Also, in the resultant states the hot carriers are thermalized, which does not match with what they claimed (although it requires further clarification)

Other minor issues:

1) In line 300, the authors stated "Our experiment results reveal the strong short-range electron-TO coupling in MXenes". I am wondering how to define "short" or "long" coupling exactly? Can the author provide any explanation?

2) In line 85, electron-photon coupling should be electron-phonon coupling;

3) In line 229, we discussion >>> we discuss;

4) In line 295, Mo2S >>> MoS2;

Reviewer #2 (Remarks to the Author):

This paper investigates the phonon and electron dynamics in MXenes. The paper is interesting and this study may be open new avenues for MXenes that having a great impact the Materials Science.

The authors mentioned that the injection of photons into MXene by pump light promote somehow atomic displacement. Meanwhile the statement is well-understood. I wonder if they have any idea about the shifts in the atomic displacements. Could it affect to the stability of the MXene? In other words, the dynamic stability is affected by photons?

The authors mention "The Ti₃C₂T_x with metallic electronic structure has a lot of free electrons near the Fermi Surface". Bare Ti₃C₂ MXene is metallic. However, the termination Tx can change the electronic behavior inducing semiconducting behavior. In principle, semiconducting MXenes have not electrons near the Fermi Surface. Therefore, how does it affects to the electron-phonon coupling? If I am right, I suggest to the reviewers give briefly further details about this.

The authors investigate the DOS using DFT calculations. I did not see clearly the density functional employed. I believe that PBE was employed. If so, I recommend to give accurate results using hybrid functionals. Thus, the DOS will be more reliable overcoming the standard DFT limitations.

Reviewer #3 (Remarks to the Author):

The authors investigate electron-phonon coupling in two-dimensional transition metal carbide materials ("MXenes"). For this purpose, they employ impulsive vibrational spectroscopy and identify vibrational modes via Fourier transformation of oscillatory transient absorption data.

The scientific question addressed in this manuscript is of fundamental significance for understanding the optoelectronic properties of MXenes. While electronic or vibrational properties, taken individually, are routinely accessed via linear absorption or Raman spectroscopy, respectively, their coupling is more difficult to analyze. The data shown in the present work is of very good quality, with a signal-to-noise ratio high enough to warrant clean separation of observed transients into decaying and oscillatory components that are in turn used to provide information on the excited vibrations. The analysis is mostly convincing, and thus the work is in principle well suited for publication in Nature Communication, given the relevance of 2D materials.

There are, however, also some issues that should be addressed:

Line 150: The authors state: "The results (Fig. 3c&d) display that the electron relaxation slows down gradually with increasing the pump fluence, which is due to the thermal electrons generated following the electron electron scattering." This explanation is not very clear. Why does it slow down? Intensity-dependent dynamics have been observed in many different systems in various contexts for decades using femtosecond spectroscopy. Generally, such behavior can lead to erroneous assignment of kinetic time constants if excitation fluences are chosen too high. These higher-order effects are difficult to analyze correctly. The customary experimental procedure is therefore to avoid the high-intensity artefacts by attenuating the excitation density to such a low level that no more changes in the kinetics are observed upon lowering the intensity even more. In the displayed data, however, there is still a significant change in the kinetics between the lowest and the second-lowest fluence level, suggesting that one might need to lower the intensity even further to suppress the higher-order effects completely. Can the authors comment on this? Why did they not attenuate more? Can the authors also explain better what the origin is of the kinetic change?

Line 181: The authors state that the "electrons are strongly coupled with the A1g mode (~60 fs, 85-100%) and weakly coupled with the Eg mode (1-2 ps, 0-15 %)." Is it possible to quantify the coupling constant beyond the mere (qualitative) fact that oscillations appear more or less strongly? Electron-phonon coupling was recently quantified experimentally in another 2D material, i.e., a value was derived for the coupling constant, and strong coupling was found to the A1' phonon mode as well [D. Li et al., Nat. Commun. 12, 954 (2021)]. It might be interesting to compare the present results and discuss them in context.

Line 264: The authors write: "Secondly, the electron-phonon coupling is related to the electron density of states near the Fermi surface and the matching between electron and phonon energy levels." This statement is not straightforward to understand. Can the authors elaborate more? What does this "relation" mean specifically? Can the coupling be understood in a Franck-Condon picture as a displacement of the potential energy surface minima along the phonon coordinate?

Line 368: In the analysis, it seems that individual-wavelength oscillations are fitted. Is that correct? There exists also the possibility to employ global fitting (of all wavelengths) of kinetic and oscillatory data as shown in the literature. This could provide more information and reveal any possible variations with respect to wavelength. Can the authors comment on why they used single-wavelength analysis only, or maybe could they supplement their analysis with global fitting?

There is quite a large number of language errors in the text. Some of them are only minor, but others make the intended meaning difficult to understand. Only a few exemplary ones are listed below. There are more, and it would help if the authors read through their manuscript critically to remove them.

Line 42: There seems to be a noun missing after "nonthermal electron".

Line 229: Grammar is wrong in "Based on the above results, we discussion for".

Line 286: A verb is missing from this sentence.

Line 292: It should be "originate", not "originated".

Tobias Brixner

Response to reviewer's comments

Reviewer #1 (Remarks to the Author):

Electron-phonon interaction is essential for understanding the hot carrier cooling mechanism in solid states materials. Plasmonic MXenes aroused researchers' interest recently due to the unique electronic and phononic properties. In this work, the authors attempted to unveil the coupling between electron and phonon in MXenes by tracking the energy flow from charges to phonons, employing ultrafast impulsive vibrational spectroscopy. They found the energy of photoinduced electrons is transferred to coherent phonons without/before involving electron-electron scattering, where the electrons are strongly coupled with the A_{1g} mode (~60 fs) while weakly coupled with the E_g mode (1-2 ps).

I considered this work is systematic and provides a novel aspect to advance our current understanding about the electron-phonon interaction in MXenes. However, the current version of the manuscript has several unclear points or unsupported claims. The authors should clarify them, before the work can be considered for publication in Nature Communications.

Author Reply: We appreciate the reviewer's positive comments. Based on the reviewer's comments, we have performed substantial new experiments and analysis to improve our manuscript as follows.

Q1.1: The authors proposed the seemingly debate whether strong or weak electron-phonon coupling is in MXenes, based on the previous works. [ACS Nano 15, 14071-14079 (2021); Nat. Phys. 1-7 (2022).] However, in the case of metallic $Ti_3C_2T_x$, M. Volkov et. al. investigated the photoexcited charge carrier (with an excess energy, i.e., hot carriers) interacting with the phonons (in a non-equilibrium situation), while W. Zheng et. al. studied the charge carrier (i.e., without photoexcitation, i.e., cold carriers) scattering with the phonons (in the equilibrium condition). Note that the interaction process is basically different between phonons and (hot or cold) charge carriers. Therefore, I doubt if it makes sense to propose such a "debate".

Author Reply: Thank you for your comments. We absolutely agree with the reviewer that the interaction process is basically different between phonons and (hot or cold) charge carriers. According to the previous studies, the electron-phonon coupling strengths with increasing electron temperature is nearly constant in noble metal within $T_e < 2000K$,^{1, 2, 3} but are changed in transition metal. The recent theoretical calculation showed that electron-phonon coupling constants are reduced due to both the nonequilibrium phonon and the electrons⁴. According to the reviewer's suggestion, we have revised the sentences to "These interesting observations reported in the previous works prompt us to reinvestigate both the phonon and electron dynamics and to unravel the new energy transfer channel."

Q1.2: On the same note: The authors did not come back to this "debate" in the end. Are their results in line with one of the two papers discussed?

Author Reply: Thank you for your comments. We have removed the word "debate". A discussion comparing the results from this work and the previous two papers is added in the main text in line 312. "The previous study using terahertz spectroscopy⁵ suggested the weak long-range electron-LO coupling whereas the ultrafast electron diffraction experiment⁶ purported the strong electron-phonon coupling without clarifying the energy transfer pathway and the coupled phonon modes, which has

been clearly presented in this work.”

Q 2.1 It is an interesting observation that some specific modes are coupled to the photogenerated charge carriers in the early time scale. While this does provide a good indication for e-ph coupling, it does not provide information on the energy branching into electron-electron scattering and population of coherent phonons by non-thermalized hot carriers. From this perspective, these two energy relaxations are competing in the sub-100 fs: likely both are active and plausible. I don't follow how the author can reach a conclusion to exclude e-e scattering by stating that “our results demonstrate a new energy damping channel, in which the $\text{Ti}_3\text{C}_2\text{T}_x$ plasmonic electron energy transfers to coherent phonons (CPs) by nonthermal electron after Landau damping, without involving electron-electron scattering.”

Author Reply: Thank you for your comments. We absolutely agree with your concern that these two energy relaxations (the e-p coupling and e-e scattering) are competing in the sub-100 fs. We apologize for the inaccurate expression in the previous version. We have provided information for the energy branching into electron-electron scattering and the population of coherent phonons by non-thermalized hot carriers in main text lines 150, 171, and the detailed methods of obtaining time constants of electron-electron scattering are shown in Supplementary Note 2:

“We determined the electron-electron scattering constant half-time of ~50 fs (The details of acquiring the half-time of are described in Supplementary Fig. 7 and 8 and Supplementary Notes 2).”

“It is also noted that the electron-phonon coupling time constants are close to the electron-electron scattering time constants (~50 fs), suggesting that the two physical processes occur simultaneously. Based on the time constants, almost half of the nonthermal electrons directly couple with the CP.”

Supplementary Figure 7. (a) An UV-visible absorption spectrum of the GNRs in an aqueous solution exhibits a longitudinal SP absorption peak at ~720 nm. And an inset shows a TEM image of GNRs. (b) Transient spectra of GNR at indicated waiting times after 750 nm excitation.

Supplementary Figure 8. The time constant of electron-electron scattering. (a) Transient spectra of $\text{Ti}_3\text{C}_2\text{T}_x$ film at indicated waiting times after 790 nm excitation. (b) Transient spectra of GNRs in aqueous solution at indicated waiting times after 750 nm excitation. (c) The dynamic evolution of $\text{Ti}_3\text{C}_2\text{T}_x$ film probed at 720 nm after 780 nm excitation (d) The rise dynamic evolution of GNRs in aqueous solution probed at 720 nm after 750 nm excitation.

“Supplementary Note 2: The time constants of electron-electron scattering in plasmonic materials.

To provide information on the energy branching into electron-electron scattering and population of coherent phonons by non-thermalized electrons, we obtained the electron-electron scattering half-time of ~ 50 fs in $\text{Ti}_3\text{C}_2\text{T}_x$ and ~ 125 fs in gold nanorods (GNRs, characterizations as shown in Supplementary Figs.7a) from transient absorption (TA) spectra as shown in Supplementary Fig.8 a & b. The methods to achieve the above values are as follows:

The pump-probe technique investigates electron-electron scattering using a laser pulse to excite electrons and measure the optical response in plasmonic materials. After absorbing photons, the electron temperature increases *via* electron-electron scattering to change the dielectric function, resulting in the SP band to blueshift and broaden within ~ 100 fs. Then, the SP band shifts back and becomes narrow on the ps-ns timescale as the electron temperature cools down^{2, 7}. The observed time-dependent transient spectra feature of GNRs (Supplementary Fig.7b) are similar with the previous reports^{2, 7}. Hence, the electron-electron scattering time could be experimentally obtained by observing the blue-shifted time of the SP band. In our observation (supplementary Fig.8 a & b), the plasmonic bands in TA spectra showed that the blue-shifted times are ~ 100 and ~ 250 fs in $\text{Ti}_3\text{C}_2\text{T}_x$ and GNRs, respectively, indicating the electron-electron scattering half-times of ~ 50 fs in $\text{Ti}_3\text{C}_2\text{T}_x$ and ~ 125 fs in GNRs. In addition, another method to get a time constant of electron-electron scattering is fitting a rise dynamic curve at SP band^{1, 8}. Supplementary Fig.8 c and d showed the rise time of 100 ± 15 fs in $\text{Ti}_3\text{C}_2\text{T}_x$ and 110 ± 20 fs in GNRs. Interesting, the obtained half-times (~ 125 and ~ 50 fs) from the blue-shifted features of TA spectra and rise dynamic time (110 ± 20 and 100 ± 15 fs) are nearly consistent in GNRs, but are different in $\text{Ti}_3\text{C}_2\text{T}_x$. For $\text{Ti}_3\text{C}_2\text{T}_x$, analysis from the blue-shifted feature of TA spectral is more accurate than the rise dynamic fit. The reason for the difference is that the GBS and EAS in TA spectral are overlapped. Hence, the timescales between electron-electron scattering (~ 50 fs) and electron-phonon coupling (57 ± 10 and 39 ± 10 fs) are close. And the energy branching should be close in the two physical processes.

Q 2.2 The author aimed to support their claim by fluence dependent studies in a rather indirect way. They claimed that “the formation time and phase of CP (Fig. 3c&d and Supplementary Fig. 7 and 8; marked lines showing no obvious phase shift in CP) are independent of the pump fluence, indicating that the CP stems from nonthermal electrons rather than thermal electrons.” I don't follow the argument. In the "linear" excitation regime, the observables or lifetimes can be independent of the fluence. So please clarify it, and this is a very strong claim which requires further support: ideally a control study in a system where thermalization is known to be the main pathway for energy relaxation and compare the data to the current study.

Author Reply: Thank you for your comments. We have added a control experiment of gold nanorods (GNRs), in which the energy relaxation is assigned to channel I in Fig.1. Please see the main text line 160 and Supplementary Note 3.

We agreed with the reviewer's concern about “In the "linear" excitation regime, the observables or lifetimes can be independent of the fluence” for semiconductor or molecules, and even semiconductor (Mo_2CT_x , see Supplementary Fig. 24, the dynamic traces of the independent pump-fluence). But for metals, generally, the lifetimes are dependent of the fluence since the electron heat capacity increases linearly with temperature in metals, which are significantly different from semiconductors and molecules. In metal, the e–p coupling time without thermal electron can be obtained by the linear fit of the pump fluence-dependent e–p coupling time constants to get the intercept. We take an example to support this issue, see below in Fig.1 and 2 adopted from the previous study (Nature Communications 7, 13240 (2016)).

Pump power-dependent electron dynamics in the **metallic** Au-940 and Au-520 (Fig. 1a and b) showed obvious changes in the electron relaxation time with increasing pump fluence, but the electron relaxation constant of the **non-metallicity** Au cluster (Fig.1d) does not change significantly. The electron-phonon coupling time constant without thermal electron can be obtained from the intercepts (Fig. 2 a, b) by linear fitting pump fluence electron relaxation times. This method is widely used in metals^{9, 10, 11}. The experimentally measured electron-phonon interaction time constant increases with increasing electron temperature via electron-electron scattering^{9, 11, 12, 13}, which can be understood with the two-temperature model^{14, 15, 16} as follows,

Fig.1 Pump power-dependent electron dynamics of gold nanoparticles. Normalized decay kinetics

as a function of pump fluence for (a) Au-₉₄₀, (b) Au-₅₂₀, (c) Au₃₃₃ and (d) Au₁₄₄; the kinetics at the maximum of SPR peak of each gold nanoparticle was monitored; for Au₁₄₄, GSB ~460 nm was monitored. The data is from Nature Communications 7, 13240 (2016).

Fig.2 (a) The extracted electron–phonon coupling time constants of Au 13 nm NPs, Au-₉₄₀, Au-₅₂₀ and Au₃₃₃ as a function of pump fluence. (b) The extracted electron–phonon coupling time constants for Au₃₃₃ and Au₁₄₄ as a function of pump fluence. The data is from Nature Communications 7, 13240 (2016).

with increasing electron temperature via electron–electron scattering and enhance fluence of pump pulse in metal^{9, 11, 12, 13}, which can be understood with the two-temperature model^{14, 15, 16} as follows.

$$C_e(T_e) \frac{dT_e}{dt} = -g(T_e - T_l)$$

$$C_l \frac{dT_l}{dt} = g(T_e - T_l)$$

T_e and T_l are the temperatures of electron and lattice vibration (phonon), C_e and C_l are the electronic and lattice heat capacities. g is the electron-phonon coupling constant. Since the electron heat capacity increases linearly with temperature in metal, $C_e(T_e) = \gamma T_e$, γ is the electron heat capacity constant, thermal electron relaxation is dependent on the fluence of pump pulse. And thermal electron relaxation time constant is equivalent to the time of the thermal electron to coupling the lattice. Thus, the phonon dynamic should be changed accompanied with thermal electron relaxation^{9, 10, 11}. Besides, the result in Supplementary Fig. 21 (i) shown the pump fluence-dependent of transient signal probed at 910 nm, and the inset exhibits the data of signal size (circles) at peak and a linear fitting result (line).

We added this part in the main text line 160: “This interpretation is confirmed by a control experiment employing the sample of gold nanorods (GNRs), in which the dynamic process is unambiguously assigned to the channel I (Fig.1). The oscillating dynamic traces of GNRs (see Supplementary Fig. 11) exhibit the obvious phase shift and the CP dynamics variation with increasing the pump fluence (see Supplementary Note 3).”

Supplementary Figure 11. The dynamics of GNRs in aqueous solution at probe 720 nm after 750 nm excitation with fitted by two-exponential function. (a) Pump-probe signals were monitored with low and high pump fluences and insert is magnified area of 10-30 ps **(b)** Low pump fluence with 30 nJ/pulse. **(c)** High pump fluence with 120 nJ/pulse. **(d)** The phonon dynamics was obtained by fitting data. The solid line is a fit to the electron dynamics and the dashed line is fit to obtain the phonon dynamics. The circles are experimental data.

“Supplementary Note 3: The pump fluence-dependent dynamic evolution of GNRs.

To further confirm the channel III in Fig 1 in $\text{Ti}_3\text{C}_2\text{T}_x$, a control experiment of GNRs was performed. The pump fluence-dependent dynamics in GNRs were monitored as shown in Supplementary Fig.11 a and the inset. The dynamics showed significant phase shift and delayed appearance time ($\Delta T \approx 5$ ps) of CP signal with increasing the pump fluence. The supplementary Fig.11 b and c showed different relaxation dynamics after excitation with a low and high pump fluence, which could be fitted with a two-exponential function. The fitting results exhibited different decay time constants (2.5 ± 0.2 and 5.1 ± 0.3 ps). These timescales are commensurate with electron-phonon coupling in gold nanostructures^{9,13}.

The dynamics of GNRs could be understood with the two-temperature model^{14, 15, 16}. The equations are given as follows.

$$C_e(T_e) \frac{dT_e}{dt} = -g(T_e - T_l) \quad (1)$$

$$C_l \frac{dT_l}{dt} = g(T_e - T_l) \quad (2)$$

T_e and T_l are the temperatures of electron and lattice vibration (phonon), C_e and C_l are the electronic and lattice heat capacities. g is the electron-phonon coupling constant. The electron decay-rate (dT_e/dt) is consistent with the phonon rise-rate (dT_l/dt). Since the electron heat capacity ($C_e(T_e) = \gamma T_e$, the γ is the electron heat capacity constant) depends on electron temperature, the measured electron-phonon interaction time constants in our experiment (Supplementary Fig.11 b and c) should accompany with increasing electron temperature after electron-electron scattering and pump-fluence, consistent with these previous observations^{9, 11, 12, 13}. Meanwhile, the phonon rise-time

constant value is also higher with increasing pump fluence. Supplementary Fig.11 d showed the two fitted phonon dynamics results with two pump fluences, indicating that the phonon dynamic traces reach their maximum at ~12 and ~17 ps after excitation with low and high pump fluence, respectively. The difference ($\Delta T \approx 5$ ps) between the two times (12 and 17 ps) is consistent with the delayed appearance time ($\Delta T \approx 5$ ps) of the CP signal in Supplementary Fig.11 a (inset). The dynamic process in the GRNs corresponds to thermal electron-phonon coupling (The Channel I in Fig 1).”

Q3 In line 181, the authors stated that the electrons are strongly coupled with the specific phonon modes. By comparing the peaks of the phonon modes from Raman spectrum (Figure 3 h) and IVS (Supplementary Fig. 9), one would expect the frequency-shift of the modes due to the strong electron-phonon coupling. However, the spectra seem to overlap well. This applies for time-independent IVS spectra (Supplementary Fig. 9). The discussion should be included in the revised ms.

Author Reply: Thank you for your comments. The frequency-shift of the modes has not been observed, which may be due to the limitation of the time resolution (~40 fs) for our experimental system. In our experiments, we have not observed the significant frequency-shift of the A_{1g} mode in $Ti_3C_2T_x$. This is similar to the single-layer $MoSe_2$ of strong electron- A'_1 phonon coupling¹⁷ and the $NiPS_3$ of the strong electron- A_{1g} phonon coupling. We have added the discussion about the frequency-shift of the modes in the main text. “The frequency-shift of the vibrational mode wasn’t observed. This may be caused by the limitation of time resolution (~40 fs) in our experimental system. Additionally, in our experiment, the significant frequency-shift of A_{1g} mode wasn’t observed in $Ti_3C_2T_x$. The result is similar to the single-layer $MoSe_2$ of strong electron- A'_1 phonon coupling¹⁷ and the $NiPS_3$ of the strong electron- A_{1g} phonon coupling.”

Q4 In Figure 2d, what is the nature of the electron relaxation process? The lifetime in this work is much shorter than that reported in the previous work. [J. Phys. Chem. C 2020, 124, 11, 6441–6447] Can the authors provide any explanations?

Author Reply: Thank you for your comments. We apologize for the no clear description. In the paper of J. Phys. Chem. C 2020, 124, 11, 6441–6447, we mainly focused on the dynamic process after the electron-coherent phonon coupling in multilayer MXene by recording the spectrum with step-length of 50-100 fs. The multilayer MXene presented strong modulation signals at near zero time and disturbed by the light scattering, which prevented us to get high-quality data at near zero time. In this paper, we mainly studied the dynamic process during the electron-coherent phonon coupling. we used single-layer MXene as the sample in which the artificial modulation signals at near zero time and the light scattering were significantly reduced.

Actually, based on the two papers, most excited electrons will decay via the electron-CP (optical phonons or hot phonons) coupling and electron-electron scattering within ~100 fs. Then, a small fraction of electrons decay via the electron-acoustic phonon coupling in 1-2 ps, and electrons and phonons reach a common temperature. The energy flow pathways could be understood by the three-temperature model¹⁸ (Fig. 4). Finally, the heat in $Ti_3C_2T_x$ diffuses to the surroundings after 2 ps (2-40 ps in Fig. 3c).

Fig. 3(a) The dynamics at 950 nm of the temporal evolution of the photoinduced absorption change for the MXene nanosheets in aqueous solution under excitation at 325 nm (blue), 500 nm (olive), and 780 nm (red) from J. Phys. Chem. C 2020, 124, 11, 6441–6447 (b) The dynamics at 950 nm of the temporal evolution of the photoinduced absorption change for the few layers-MXene nanosheets in aqueous solution and film under excitation at 780 nm with setting step-length of 20 fs. (c) The enlarged view (b) range from 0.1 to 40 ps.

Fig. 4 Sketch of the energy transfer during the relaxation process. Hot electrons generate hot phonons with characteristic time τ_α , while hot phonons dissipate their energy on a time scale $\tau_\beta \gg \tau_\alpha$. (Form Physical Review Letters **99**, 197001 (2007)).

Q5 It is difficult to imagine how the E_g and A_{1g} ratio are obtained (as presented in Figure 3j). How is the population (in %) estimated? In figure 3i, with the signal to noise level I don't see any difference between the three data shown. Please clarify

Author Reply: Thank you for your comments. We have added the information about how the E_g and A_{1g} ratios are obtained, please see the main text line 475.

“Branching ratios of the two modes with various near-infrared pumps extracted from the data of Fig. 3a & b and Supplementary Fig. 14”. And relative ratios of the A_{1g} and E_g modes respectively originating from $A_{1g}/(A_{1g}+E_g)$ and $E_g/(A_{1g}+E_g)$ were acquired by using formula (2) to fit. The error bars represent standard deviation.

$$S_c(t) = \sum_{i=1,2} A_i \cos(\omega_i t + \theta_i) * \left(e^{-\frac{t}{\tau_{ri}}} \right) \quad (2)$$

The fitting dynamic trace could obtain the coefficients A_1 of the A_{1g} mode and A_2 of the E_g mode. The populations of A_{1g} is $A_1/(A_1+A_2)$ and of E_g is $A_2/(A_1+A_2)$.

It is reasonable that don't see any difference between the three data shown since the electron- A_{1g} coupling is dominant and the electrons- E_g is fairly weak (not more than 15%) no matter what the excitation wavelengths are in our experiment.

Q 6 The authors should provide the meaning of error bars and how they were obtained, in particular when they compare the nonthermal electron decay time constants and coherence phonon formation time in lines 158-161.

Author Reply: Thank you for your comments. We have provided the information of error analysis in the main text line 393 and Supplementary Note 7.

“The error bars represent standard deviation. Error analysis has been shown in Supplementary Notes 7. The error bars result from the sum of fitting measured data and the experimental system errors. For example, Supplementary Fig. 19a & b show an error (9 fs) from fitted a $\text{Ti}_3\text{C}_2\text{T}_x$ dynamics with deconvoluted and an error (6 fs, sum of a peak and a width error) from fitted instrument response function. Hence the sum of errors is 15 fs.”

Supplementary Figure 19. Experimental error analysis. (a) Measured data (circle) and fitting result (solid line) at pump 780 nm probe 900 nm with deconvoluted in $\text{Ti}_3\text{C}_2\text{T}_x$. (b) A measured instrument response function (IRF) signal in a substrate of CaF_2 was fitted with a Gaussian function.

Q7 The schematics in figure 1 do not reflect the channel III that the author proposed. What it looks like is just an escape from thermalization. It does not show the coupling from nonthermal hot carriers to CP. Also, in the resultant states the hot carriers are thermalized, which does not match with what they claimed (although it requires further clarification).

Author Reply: Thank you for the valuable suggestions. Following reviewer suggestions, we have adjusted and clarified Figure 1 and add caption in the main text line 440 as follows.

Fig. 1 Schematic illustration of three SP-vibration coupling channels after optical excitation. Channel I: The SP decays to generate nonthermal electrons after Landau damping, the nonthermal electrons then evolve into thermal electrons through the electron-electron scattering, and thermal electrons relax to phonons *via* the electron-phonon coupling; Channel II: The SP directly converts into phonons; Channel III: The SP decays to generate nonthermal electrons, then the nonthermal electrons directly couple with phonons.

Other minor issues:

1) In line 300, the authors stated “Our experiment results reveal the strong short-range electron-TO coupling in MXenes”. I am wondering how to define “short” or “long” coupling exactly? Can the author provide any explanation?

Author Reply: Thank you for your comments. In general, the short-range electron-phonon coupling is in one lattice unit and the long-range electron-phonon coupling extends to several lattice units.

2) In line 85, electron-photon coupling should be electron-phonon coupling;

Author Reply: Thank you for your comments. We have revised “electron-photon coupling” to “**electron-phonon coupling**” in the main text line 83.

3) In line 229, we discussion >>> we discuss;

Author Reply: Thank you for your comments. We have revised “we discussion” to “**we discuss**” in the main text line 244.

4) In line 295, Mo2S >>> MoS2;

Author Reply: Thank you for your comments. We have revised “Mo₂S” to “**MoS₂**” in the main text line 307.

Reviewer #2 (Remarks to the Author):

This paper investigates the phonon and electron dynamics in MXenes. The paper is interesting and this study may be open new avenues for MXenes that having a great impact the Materials Science.

Q1. The authors mentioned that the injection of photons into MXene by pump light promote somehow atomic displacement. Meanwhile the statement is well-understood. I wonder if they have

any idea about the shifts in the atomic displacements. Could it affect to the stability of the MXene? In other words, the dynamic stability is affected by photons?

Author Reply: Thank you for your comments. We absolutely agree with the reviewer's concern about stability. The strong laser irradiation would affect the stability of MXene under the conditions of oxygen and water. In order to ensure that MXene was stable, in our experiment, the MXene films were placed into a vacuum cell (Oxford instruments, Optistat DN-V) with a vacuum pump (PFEIFFER HiCube 80 Pumping Station) during the record of the broadband IVS and pump-probe data. We have confirmed that the signal is very stable during the measurement. We have added this detailed description in the main text method part.

Q2.1 The authors investigate the DOS using DFT calculations. I did not see clearly the density functional employed. I believe that PBE was employed. If so, I recommend to give accurate results using hybrid functionals. Thus, the DOS will be more reliable overcoming the standard DFT limitations.

Author Reply: Thank you for your comments. Following the reviewer's suggestions, we employed the hybrid functionals to calculate the electron band structure for $\text{Ti}_3\text{C}_2\text{O}_2$ and $\text{Ti}_3\text{C}_2(\text{OH})_2$ and put the new results of $\text{Ti}_3\text{C}_2\text{O}_2$ into the main text in Fig. 4 a & b, and modified the description in lines 226, 400, 479-482.

Fig. 4 The calculated band structures and density of states (DOS) of $\text{Ti}_3\text{C}_2\text{T}_x$ ($\text{Ti}_3\text{C}_2\text{O}_2$). (a) Hybrid functional electron structures and DOS. Marked around -1.3 eV as the a band, around 0.0 eV as the b band, and about 1.6 eV as the c band (main contribution of Ti1/Ti2 d_{z^2} , d_{xz} , d_{xy} orbitals), approximately 2.6 eV as the d band (main contribution of Ti1/Ti2 d_{z^2}). (b) Hybrid functional fatband of d_{xy} , d_{xz} and d_{z^2} of Ti (The diameters of the circles represent the intensity of DOS). (c) The phonon dispersions band and DOS.

The results of employing the hybrid functionals to calculate electron structures of $\text{Ti}_3\text{C}_2\text{O}_2$ and $\text{Ti}_3\text{C}_2(\text{OH})_2$ are shown in Fig. 5 a & b. And The standard DFT calculated band structures of $\text{Ti}_3\text{C}_2\text{O}_2$ and $\text{Ti}_3\text{C}_2(\text{OH})_2$ are shown in Fig. 5 c & d. These results show $\text{Ti}_3\text{C}_2\text{O}_2$ and $\text{Ti}_3\text{C}_2(\text{OH})_2$ are metallic. Meanwhile, the hybrid functionals and the standard DFT calculation give similar results. Compared with the standard DFT calculation, the hybrid functional results show that band structure shifts slightly. Our calculated results are consistent with the previous Hybrid density functional study¹⁹.

Fig. 5 (a, b) The Hybrid functional calculated electron structures of $\text{Ti}_3\text{C}_2\text{O}_2$ and $\text{Ti}_3\text{C}_2(\text{OH})_2$, respectively. **(c, d)** The standard DFT calculated band structures of $\text{Ti}_3\text{C}_2\text{O}_2$ and $\text{Ti}_3\text{C}_2(\text{OH})_2$, respectively.

Q2.2 The authors mention “The $\text{Ti}_3\text{C}_2\text{T}_x$ with metallic electronic structure has a lot of free electrons near the Fermi Surface”. Bare Ti_3C_2 MXene is metallic. However, the termination T_x can change the electronic behavior inducing semiconducting behavior. In principle, semiconducting MXenes have not electrons near the Fermi Surface. Therefore, how does it affects to the electron-phonon coupling? If I am right, I suggest to of the reviewers give briefly further details about this.

Author Reply: Thank you for your comments. We have performed new experiments on HF-etched $\text{Ti}_3\text{C}_2\text{T}_x$. The terminal groups of $\text{Ti}_3\text{C}_2\text{T}_x$ in our experiment include F, O and OH. On the basis of the Hybrid functional calculated results shown in Fig. 5, $\text{Ti}_3\text{C}_2\text{O}_2$ and $\text{Ti}_3\text{C}_2(\text{OH})_2$ are metallic. And the calculated $\text{Ti}_3\text{C}_2\text{O}_2$, $\text{Ti}_3\text{C}_2(\text{OH})_2$ and $\text{Ti}_3\text{C}_2\text{F}_2$ also exhibit the metallic property in previous study¹⁹. According to the previous experimental result²⁰, the $\text{Ti}_3\text{C}_2\text{T}_x$ is also metallic. Therefore, the $\text{Ti}_3\text{C}_2\text{T}_x$ is metallic in our experiments.

To clarify the effects of terminal groups to the electron-phonon coupling, we performed a control experiment of HF- $\text{Ti}_3\text{C}_2\text{T}_x$ as shown in Fig.6. Fig.6a shows the XPS spectra of HF-etched $\text{Ti}_3\text{C}_2\text{T}_x$ and LiF-etched $\text{Ti}_3\text{C}_2\text{T}_x$. The results suggest the HF-etched $\text{Ti}_3\text{C}_2\text{T}_x$ has more fluorine and less oxygen terminal group than LiF-etched $\text{Ti}_3\text{C}_2\text{T}_x$, which are consistent with the previous report²¹. Fig.6b shows dynamic traces of the HF-etched $\text{Ti}_3\text{C}_2\text{T}_x$. Fig.6c shows a comparison of pump fluence-dependent electron-phonon coupling time constants between the HF-etched $\text{Ti}_3\text{C}_2\text{T}_x$ and LiF-etched $\text{Ti}_3\text{C}_2\text{T}_x$. The intercepts in Fig.6c indicate the intrinsic electron-phonon coupling time constants of HF-etched $\text{Ti}_3\text{C}_2\text{T}_x$ (~42 fs) and LiF-etched $\text{Ti}_3\text{C}_2\text{T}_x$ (~39 fs), suggesting a slight influence on time constants of electron-phonon coupling by changing from oxygen to fluorine

terminal groups.

Fig. 6 (a) Comparison of XPS spectra between the HF-etched $\text{Ti}_3\text{C}_2\text{T}_x$ and LiF-etched $\text{Ti}_3\text{C}_2\text{T}_x$. (b) 780nm pump fluence-dependent dynamics data of HF-etched $\text{Ti}_3\text{C}_2\text{T}_x$ were monitored at 900 nm. (c) Comparison of pump fluence-dependent electron-phonon coupling time constants between the HF-etched $\text{Ti}_3\text{C}_2\text{T}_x$ and LiF-etched $\text{Ti}_3\text{C}_2\text{T}_x$.

Reviewer #3 (Remarks to the Author):

The authors investigate electron-phonon coupling in two-dimensional transition metal carbide materials ("MXenes"). For this purpose, they employ impulsive vibrational spectroscopy and identify vibrational modes via Fourier transformation of oscillatory transient absorption data. The scientific question addressed in this manuscript is of fundamental significance for understanding the optoelectronic properties of MXenes. While electronic or vibrational properties, taken individually, are routinely accessed via linear absorption or Raman spectroscopy, respectively, their coupling is more difficult to analyze. The data shown in the present work is of very good quality, with a signal-to-noise ratio high enough to warrant clean separation of observed transients into decaying and oscillatory components that are in turn used to provide information on the excited vibrations. The analysis is mostly convincing, and thus the work is in principle well suited for publication in Nature Communication, given the relevance of 2D materials. There are, however, also some issues that should be addressed:

Q1 Line 150: The authors state: "The results (Fig. 3c&d) display that the electron relaxation slows down gradually with increasing the pump fluence, which is due to the thermal electrons generated following the electron electron scattering." This explanation is not very clear. Why does it slow down? Intensity-dependent dynamics have been observed in many different systems in various contexts for decades using femtosecond spectroscopy. Generally, such behavior can lead to erroneous assignment of kinetic time constants if excitation fluences are chosen too high. These higher-order effects are difficult to analyze correctly. The customary experimental procedure is therefore to avoid the high-intensity artefacts by attenuating the excitation density to such a low level that no more changes in the kinetics are observed upon lowering the intensity even more. In the displayed data, however, there is still a significant change in the kinetics between the lowest and the second-lowest fluence level, suggesting that one might need to lower the intensity even further to suppress the higher-order effects completely. Can the authors comment on this? Why did they not attenuate more? Can the authors also explain better what the origin is of the kinetic change?

Author Reply: Thank you for your comments. We apologized for the no detailed discussion about

the pump fluence experiment in the previous version. Following reviewer suggestions, we have added the detailed discussion about the pump fluence experiment in the main text line 154 and Supplementary Note 3.

We agree with the reviewer's concern about "Generally, such behavior can lead to erroneous assignment of kinetic time constants if excitation fluences are chosen too high", for semiconductor, molecules, and even semimetal (such as Mo_2CT_x , see Supplementary Fig. 24, dynamic traces of the independent pump-fluence). But for metal, generally, the electronic lifetimes are dependent on the fluence since the electron heat capacity increases linearly with temperature in metal, which is significantly different from semiconductors or molecules. Note that, in general, the electron-phonon time constants increase after electron-electron scattering in metals. The e-p coupling time constant without thermal electron can be obtained by the linear fit of the pump fluence-dependent e-p coupling time constants to get the intercept. More detail can be seen in the **Author Reply of review 1, Q2**.

Q2 Line 181: The authors state that the "electrons are strongly coupled with the A_{1g} mode (~60 fs, 85-100%) and weakly coupled with the E_g mode (1-2 ps, 0-15 %)." Is it possible to quantify the coupling constant beyond the mere (qualitative) fact that oscillations appear more or less strongly? Electron-phonon coupling was recently quantified experimentally in another 2D material, i.e., a value was derived for the coupling constant, and strong coupling was found to the A_1' phonon mode as well [D. Li et al., Nat. Commun. 12, 954 (2021)]. It might be interesting to compare the present results and discuss them in context.

Author Reply: Thank you for your comments. Your suggestion is very helpful for us. The Huang-Rhys could evaluate the electron-phonon coupling strength in semiconductors. Recently, in D. Li et al., Nat. Commun. 12, 954 (2021), the Huang-Rhys of the semiconductor MoSe_2 was determined by high quality experimental measurement. However, the $\text{Ti}_3\text{C}_2\text{T}_x$ is metallic with overlaps between the valence and conduction bands, and the excited states of $\text{Ti}_3\text{C}_2\text{T}_x$ are continuous, different from 2D semiconductors with a band gap and discrete resonance states. Hence it is hard to observe discrete resonance state or apply the two-electronic-level approximation in MXene. Alternatively, we determined the electron-phonon coupling constant (λ) in the framework of metals²² in the main text line 320 and Supplementary Note 5. The present results were compared and discussed with MoSe_2 [D. Li et al., Nat. Commun. 12, 954 (2021)] in the main text lines 307, 326 and 320 and Supplementary Note 5:

"Additionally, the significant frequency-shift of the A_{1g} mode has not been observed in $\text{Ti}_3\text{C}_2\text{T}_x$ in our experiment, which is similar to the single-layer MoSe_2 of strong electron- A_1' phonon coupling¹⁷ and the NiPS_3 of the strong electron- A_{1g} phonon coupling²³"

"Furthermore, we determined the electron-phonon coupling constant (λ) in the framework of metals²². The values of λ are 1.62 ± 0.33 for A_{1g} and 0.054 ± 0.018 for E_g in $\text{Ti}_3\text{C}_2\text{T}_x$ from the measured electron and CP dynamics. The details of obtaining λ can be found in the Supplementary Note 5. Compared with the values ($\lambda \sim 0.03-1.45$) of most metals and semi-metals, such as Au, Cu, Ti, V, Nb, W, Pb¹⁵ and Ag, Al, TiAl²⁴ and graphene²⁵, the electron-phonon coupling constant in $\text{Ti}_3\text{C}_2\text{T}_x$ indicates a strong electron- A_{1g} phonon coupling and a weak electron- E_g phonon coupling." The summarized electron-phonon coupling times and constants are shown in Table R 1.

Table R 1 The electron-phonon coupling constant in various metals.

Material	τ_{ep} (fs)	$\lambda\langle\omega^2\rangle$ (meV ²)	λ
Ti ₃ C ₂ T _x (A _{1g})	50	605	1.62
Ti ₃ C ₂ T _x (E _g)	1500	40.5	0.054
Au	1900 ^b	23 ^a	0.13 ^a
Cu	1400 ^b	29 ^a	0.08 ^a
Ti	160 ^b	350 ^a	0.58 ^a
V	170 ^b	280 ^a	0.80 ^a
Nb	170 ^b	320 ^a	1.16 ^a
W	710 ^b	110 ^a	0.26 ^a
Pb	840 ^b	45 ^a	1.45 ^a
Bi ₂ Sr ₂ CaCu ₂ O _{8+δ}	110 ^c	360 ^c	< 0.25 ^c
YBa ₂ Cu ₃ O _{6.5}	100 ^b	400 ^b	
La _{1.8} Sr _{0.15} CuO ₄	45 ^b	800 ^b	
Graphene	150 ^d		0.033 ^d
Ag			0.13 ^e
Al			0.43 ^e
Pt			0.58 ^e
CuAu			0.31 ^e
Cu ₃ Au			0.34 ^e
TiAl			0.57 ^e

^a Reference¹⁵

^b Reference(supporting material)²⁶

^c Reference¹⁸

^d Reference²⁵

^e Reference²⁴

“Supplementary Note 5: The calculation the electron–phonon coupling constant λ .

In order to compare the electron-phonon coupling strengths with other materials, we determined the electron-phonon coupling factors and constants, $\lambda\langle\omega^2\rangle$ and λ ²² as follows:

$$\lambda\langle\omega^2\rangle = 605 \text{ meV}^2 \text{ and } \lambda = 1.62 \text{ (A}_{1g}\text{)};$$

$$\lambda\langle\omega^2\rangle = 20 \text{ meV}^2 \text{ and } \lambda = 0.054 \text{ (E}_g\text{)}.$$

These values were calculated by combing with the two-temperature model¹⁵ and nonequilibrium model²⁶. The electron-phonon coupling strengths are calculated as follows:

According to the two-temperature model(TTM)^{15, 26}:

$$\lambda\langle\omega^2\rangle = \frac{\pi}{3} \frac{k_B T_e}{\hbar\tau_{e-ph}} \quad (3)$$

And the nonequilibrium model (NEM)^{26, 27}:

$$\lambda\langle\omega^2\rangle = \frac{2\pi}{3} \frac{k_B T_l}{\hbar\tau_{e-ph}} \quad (4)$$

T_e is electron temperature. K_B is Boltzmann's constant, \hbar is Planck's constant, and τ_{e-p} is the electron-phonon coupling time constant. We take the average value of electron-phonon coupling time constant for two excitation wavelengths (39 fs with excitation at 780 nm and 57 fs with excitation at pump 532 nm, Fig. 3 c & d) τ_{e-ph} as 50±10 fs for A_{1g} and 1500±500 fs for the E_g (The time constant

is 1-2 ps for E_g . Hence, we take the intermediate value of 1.5 ps to calculate the coupling strength).

The values of electron-phonon coupling strength ($\lambda\langle\omega^2\rangle$) were obtained as follows:

The TTM model: $\lambda\langle\omega^2\rangle = 403 \text{ meV}^2 (A_{1g})$;

The NEM model: $\lambda\langle\omega^2\rangle = 806 \text{ meV}^2 (A_{1g})$;

The TTM model: $\lambda\langle\omega^2\rangle = 13.5 \text{ meV}^2 (E_g)$;

The NEM model: $\lambda\langle\omega^2\rangle = 27 \text{ meV}^2 (E_g)$.

The TTM is based on the assumption that an electron-electron scattering time constant (τ_{e-e}) is much shorter than an electron-phonon coupling time constant (τ_{e-ph}) while the NEM model is applicable when $\tau_{e-e} > \tau_{e-ph}$. However, the time constants of τ_{e-e} and τ_{e-ph} are quite close in $\text{Ti}_3\text{C}_2\text{T}_x$. Therefore, we take the average value of $\lambda\langle\omega^2\rangle$ in the TTM and the NEM model as $605 \text{ meV}^2 (A_{1g})$ and $20 \text{ meV}^2 (E_g)$. The $\langle\omega^2\rangle$ value could use the approximation $\langle\omega^2\rangle = \theta_D^2/2$.³ The θ_D is Debye temperature, which is 317 K in $\text{Ti}_3\text{C}_2\text{T}_x$.²⁸ The $\theta_D^2/2$ is 373 meV^2 . The calculated electron-phonon coupling constants (λ) are $\sim 1.62 (A_{1g})$ and $\sim 0.054 (E_g)$.

Q3. Line 264: The authors write: "Secondly, the electron-phonon coupling is related to the electron density of states near the Fermi surface and the matching between electron and phonon energy levels." This statement is not straightforward to understand. Can the authors elaborate more? What does this "relation" mean specifically? Can the coupling be understood in a Franck-Condon picture as a displacement of the potential energy surface minima along the phonon coordinate?

Author Reply: Thank you for the comments. We apologized for the no detailed discussion about it. In metals, basis on fermi's "golden rule", an approximate description for the electron-phonon coupling rate have been reported^{24, 29}:

$$\frac{1}{\tau_{ep}} \propto \sum_{\mathbf{k}, \mathbf{m}, \mathbf{n}} |g_{\mathbf{n}\mathbf{k}+\mathbf{q}, \mathbf{m}\mathbf{k}}|^2 \frac{\partial f(\varepsilon_{\mathbf{m}\mathbf{k}})}{\partial \varepsilon} \delta(\varepsilon_{\mathbf{m}\mathbf{k}} - \varepsilon_{\mathbf{n}\mathbf{k}+\mathbf{q}} + \hbar\omega_{\mathbf{q}\mathbf{v}})\omega_{\mathbf{q}\mathbf{v}}$$

τ_{ep} is electron-phonon coupling time constant. $|g_{\mathbf{n}\mathbf{k}+\mathbf{q}, \mathbf{m}\mathbf{k}}|^2$ is the electron-phonon coupling matrix, $f(\varepsilon_{\mathbf{m}\mathbf{k}})$ is the Dirac-Fermi distribution, $\varepsilon_{\mathbf{m}\mathbf{k}+\mathbf{q}}$ and $\varepsilon_{\mathbf{n}\mathbf{k}}$ are the electron energy levels, and $\omega_{\mathbf{q}\mathbf{v}}$ is the phonon energy level.

According to this equation, it can help us to qualitative understanding the strength of electron-phonon coupling in MXene. The strong e-p coupling rate generally comes from three conditions: (I) High phonon frequency and matching between electron and phonon energy levels. If the 2D material holds high-frequencies phonon and the energy matching between electron and phonon energy levels, the $\delta(\varepsilon_{\mathbf{m}\mathbf{k}} - \varepsilon_{\mathbf{n}\mathbf{k}+\mathbf{q}} + \hbar\omega_{\mathbf{q}\mathbf{v}})\omega_{\mathbf{q}\mathbf{v}}$ will be larger. (II) High electron density of states at the Fermi surface, $\frac{\partial f(\varepsilon_{\mathbf{m}\mathbf{k}})}{\partial \varepsilon}$, also results in a stronger e-p interaction (III) $|g_{\mathbf{n}\mathbf{k}+\mathbf{q}, \mathbf{m}\mathbf{k}}|^2$: Large e-p coupling matrix element.

We agree with the reviewer's concern that "the e-p coupling in $\text{Ti}_3\text{C}_2\text{T}_x$ can be understood in a Franck-Condon picture as a displacement of the potential energy surface minima along the phonon coordinate". However, it is difficult to determine the potential energy surface due to the complex band structure in $\text{Ti}_3\text{C}_2\text{T}_x$. $\text{Ti}_3\text{C}_2\text{T}_x$ is metallic with overlaps between the valence band and conduction bands without a band gap, and the excited states of $\text{Ti}_3\text{C}_2\text{T}_x$ are continuous, different from 2D semiconductors with a band gap and discrete resonance states. Hence it is hard to observe discrete resonance state or apply the two-electronic-level approximation in MXene.

Q4, Line 368: In the analysis, it seems that individual-wavelength oscillations are fitted. Is that correct? There exists also the possibility to employ global fitting (of all wavelengths) of kinetic and oscillatory data as shown in the literature. This could provide more information and reveal any possible variations with respect to wavelength. Can the authors comment on why they used single-wavelength analysis only, or maybe could they supplement their analysis with global fitting?

Author Reply: Thank you for your comments. Following your suggestions, we have added the global analyses³⁰ of kinetic for $\text{Ti}_3\text{C}_2\text{T}_x$. The results are shown in the main text line 394 and supplementary information line 140, and the fitting multi-wavelengths analysis oscillatory data are shown in supplementary information line 157.

We agree with the reviewer's comments. Employing global fitting could provide more information and reveal any possible variations with respect to wavelength. We used the single-wavelength analysis because the spectral response at ~900 nm has high signal intensity and small overlap between a negative bleaching and excited-state absorptions (the details of the analysis are described in Supplementary Note 1). We think probing at ~900 nm could represent the whole dynamic features, but employing global fitting is more reasonable and accurate. We have employed global fitting to our kinetic data, as shown in Supplementary Figure 21. The comparison of results between the Global fits and the single-wavelength fits displays the similarity.

We have also tried to use the programs of femtoTools package collection from Phys. Chem. Chem. Phys. **18**, 33287-33302 (2016) to analyze the oscillatory data, but it is very difficult for us to use the programs for our data analysis. Alternatively, we have fitted multi-wavelengths to analyze oscillatory data and acquired the formation and relaxation time constants, as shown in Supplementary Fig. 22. The results are similar to the single-wavelength analysis (55 ± 20 fs and 680 ± 42 fs).

Supplementary Figure 21. The global fits for the $\text{Ti}_3\text{C}_2\text{T}_x$ film under the excitation of 780 nm. (a-e) Acquired decay-associated spectra (DAS) from the global fits on the transient data at different pump fluences. The absence of spectra ranging from 760 to 860 nm is attributed to the reason that the fundamental pulse leads to strong noise. (f) Selected dynamics trace monitored at indicated

wavelengths and corresponding fitting results. (g) the normalized pump fluence-dependent dynamics data (circle) probed at 910 nm and the corresponding global fitting results (lines). (h) Comparison of the fitted time constants (τ_1) using the global and single-wavelength analysis. The error bars represent standard deviation. (i) The pump fluence-dependent of transient signal probed at 910 nm, and the inset exhibits the data of signal size (circles) at peak and a linear fitting result (line).

Supplementary Figure 22. Analysis of oscillatory components via fitting multi-probe-wavelengths. (a) Oscillatory data (circles) and fitting results (lines) at indicated probe wavelengths. (b) The Probe-wavelength dependent formation and relaxation time constants. (c) The distribution of formation time constants (d) The distribution of relaxation time constants.

There is quite a large number of language errors in the text. Some of them are only minor, but others make the intended meaning difficult to understand. Only a few exemplary ones are listed below. There are more, and it would help if the authors read through their manuscript critically to remove them.

Author Reply: Thank you for your suggestions. We have revised the typo.

Line 42: There seems to be a noun missing after "nonthermal electron".

Author Reply: Thank you for your suggestions. We have inserted the word "mediation" in the sentence in main text line 39: "in which the $Ti_3C_2T_x$ plasmonic electron energy transfers to coherent phonons (CPs) by nonthermal electron mediation after Landau damping."

Line 229: Grammar is wrong in "Based on the above results, we discussion for".

Author Reply: Thank you for your suggestions. We have changed "we discussion for" to "we discuss the" in the main text line 244.

Line 286: A verb is missing from this sentence.

Author Reply: Thank you for your suggestions. We have inserted the word "are" in the sentence in main text line 301: "Our results show that the electron-CP coupling strengths of $\text{Ti}_3\text{C}_2\text{T}_x$ (electron-phonon coupling time constant: ~50 fs) and Mo_2CT_x (electron-phonon coupling time constant: ~152 fs, Supplementary Fig. 15c) are stronger".

Line 292: It should be "originate", not "originated".

Author Reply: Thank you for your suggestions. We have change "originated" to "originate" in the main text line 305.

References

1. Del Fatti N, Voisin C, Achermann M, Tzortzakis S, Christofilos D, Vallée F. Nonequilibrium electron dynamics in noble metals. *Physical Review B* **61**, 16956-16966 (2000).
2. Brown AM, Sundararaman R, Narang P, Schwartzberg AM, Goddard WA, Atwater HA. Experimental and Ab Initio Ultrafast Carrier Dynamics in Plasmonic Nanoparticles. *Physical Review Letters* **118**, 087401 (2017).
3. Lin Z, Zhigilei LV, Celli V. Electron-phonon coupling and electron heat capacity of metals under conditions of strong electron-phonon nonequilibrium. *Physical Review B* **77**, 075133 (2008).
4. Miao W, Wang M. Nonequilibrium effects on the electron-phonon coupling constant in metals. *Physical Review B* **103**, 125412 (2021).
5. Zheng W, *et al.* Band transport by large Fröhlich polarons in MXenes. *Nature Physics*, (2022).
6. Volkov M, Willinger E, Kuznetsov DA, Müller CR, Fedorov A, Baum P. Photo-Switchable Nanoripples in $\text{Ti}_3\text{C}_2\text{T}_x$ MXene. *ACS Nano* **15**, 14071-14079 (2021).
7. Rotenberg N, Caspers JN, van Driel HM. Tunable ultrafast control of plasmonic coupling to gold films. *Physical Review B* **80**, 245420 (2009).
8. Voisin C, *et al.* Size-Dependent Electron-Electron Interactions in Metal Nanoparticles. *Physical Review Letters* **85**, 2200-2203 (2000).
9. Link S, El-Sayed MA. Spectral Properties and Relaxation Dynamics of Surface Plasmon Electronic Oscillations in Gold and Silver Nanodots and Nanorods. *The Journal of Physical Chemistry B* **103**, 8410-8426 (1999).
10. Hodak JH, Henglein A, Hartland GV. Electron-phonon coupling dynamics in very small (between 2 and 8 nm diameter) Au nanoparticles. *The Journal of Chemical Physics* **112**, 5942-5947 (2000).
11. Tagliabue G, *et al.* Ultrafast hot-hole injection modifies hot-electron dynamics in Au/p-GaN heterostructures. *Nature Materials* **19**, 1312-1318 (2020).
12. Park S, Pelton M, Liu M, Guyot-Sionnest P, Scherer NF. Ultrafast Resonant Dynamics of Surface Plasmons in Gold Nanorods. *The Journal of Physical Chemistry C* **111**, 116-123

- (2007).
13. Zhou M, *et al.* Evolution from the plasmon to exciton state in ligand-protected atomically precise gold nanoparticles. *Nature Communications* **7**, 13240 (2016).
 14. Schoenlein RW, Lin WZ, Fujimoto JG, Eesley GL. Femtosecond studies of nonequilibrium electronic processes in metals. *Physical Review Letters* **58**, 1680-1683 (1987).
 15. Brorson SD, *et al.* Femtosecond room-temperature measurement of the electron-phonon coupling constant in metallic superconductors. *Physical Review Letters* **64**, 2172-2175 (1990).
 16. Brorson SD, Fujimoto JG, Ippen EP. Femtosecond electronic heat-transport dynamics in thin gold films. *Physical Review Letters* **59**, 1962-1965 (1987).
 17. Li D, *et al.* Exciton-phonon coupling strength in single-layer MoSe₂ at room temperature. *Nature Communications* **12**, 954 (2021).
 18. Perfetti L, Loukakos PA, Lisowski M, Bovensiepen U, Eisaki H, Wolf M. Ultrafast Electron Relaxation in Superconducting by Time-Resolved Photoelectron Spectroscopy. *Physical Review Letters* **99**, 197001 (2007).
 19. Xie Y, Kent PRC. Hybrid density functional study of structural and electronic properties of functionalized Ti_{n+1}X_n (X=C, N) monolayers. *Physical Review B* **87**, 235441 (2013).
 20. Zheng W, *et al.* Band transport by large Fröhlich polarons in MXenes. *Nature Physics* **18**, 544-550 (2022).
 21. Hope MA, *et al.* NMR reveals the surface functionalisation of Ti₃C₂ MXene. *Physical Chemistry Chemical Physics* **18**, 5099-5102 (2016).
 22. Allen PB. Theory of thermal relaxation of electrons in metals. *Physical Review Letters* **59**, 1460-1463 (1987).
 23. Ergeçen E, *et al.* Magnetically brightened dark electron-phonon bound states in a van der Waals antiferromagnet. *Nature communications* **13**, 1-7 (2022).
 24. Tong Z, Li S, Ruan X, Bao H. Comprehensive first-principles analysis of phonon thermal conductivity and electron-phonon coupling in different metals. *Physical Review B* **100**, 144306 (2019).
 25. Johansen JC, *et al.* Direct View of Hot Carrier Dynamics in Graphene. *Physical Review Letters* **111**, 027403 (2013).
 26. Gadermaier C, *et al.* Electron-Phonon Coupling in High-Temperature Cuprate Superconductors Determined from Electron Relaxation Rates. *Physical Review Letters* **105**, 257001 (2010).
 27. Kabanov VV, Alexandrov AS. Electron relaxation in metals: Theory and exact analytical solutions. *Physical Review B* **78**, 174514 (2008).
 28. Khaledialidusti R, Anasori B, Barnoush A. Temperature-dependent mechanical properties of Ti_{n+1}C_nO₂ (n = 1, 2) MXene monolayers: a first-principles study. *Physical Chemistry Chemical Physics* **22**, 3414-3424 (2020).
 29. Huang Y, Zhou J, Wang G, Sun Z. Abnormally Strong Electron-Phonon Scattering Induced Unprecedented Reduction in Lattice Thermal Conductivity of Two-Dimensional Nb₂C. *Journal of the American Chemical Society* **141**, 8503-8508 (2019).
 30. Schott S, Röss L, Hrušák J, Nuernberger P, Brixner T. Identification of photofragmentation patterns in trihalide anions by global analysis of vibrational wavepacket dynamics in broadband transient absorption data. *Physical Chemistry Chemical Physics* **18**, 33287-

33302 (2016).

REVIEWER COMMENTS

Reviewer #1 (Remarks to the Author):

In a substantially revised ms, the authors have made much effort to clear out the most of my concerns. I have one last major comment left for the author (mostly related to figure 5b): What is the exact energy dissipation path for thermalized hot electrons? They should go somewhere in the end. Why can they not excite e.g. A_{1g} or E_g modes?

Reviewer #2 (Remarks to the Author):

In my opinion the authors addressed the concerns required and the manuscript is suitable for publication in Nature Communication

Reviewer #3 (Remarks to the Author):

The authors have provided a significantly revised version of their manuscript containing additional data and additional analysis. They have addressed all my previous concerns, and I gladly recommend publication in Nature Communications.

Response to reviewer's comments

Reviewer #1 (Remarks to the Author):

In a substantially revised ms, the authors have made much effort to clear out the most of my concerns. I have one last major comment left for the author (mostly related to figure 5b): What is the exact energy dissipation path for thermalized hot electrons? They should go somewhere in the end. Why can they not excite e.g. A_{1g} or E_g modes?

Author Reply: We appreciate your comments. We consider that the energy of thermalized electron mainly dissipates to acoustic phonons. Based on our experimental and calculated results, the thermal electron and A_{1g} mode almost reach a quasi-equilibrium temperature after electron-electron scattering and nonthermal electron- A_{1g} phonon coupling. Hence, the energy exchange between the thermal electron and A_{1g} mode is not efficient after 100 fs. However, around 100 fs, the acoustic phonons are still cold and the most of thermal electron energy could dissipate to the cold acoustic phonons. While for the thermal electron- E_g coupling, three components would determine its coupling strength: (1) low phonon density of state in the E_g mode (main text Fig. 4c), (2) low electron density of state in dxz , dxy , dyz and dx^2-y^2 orbitals near fermi level (Figure 1) and (3) small in-plane momentum of thermal electron. Thus, the energy transfer from the thermal energy to E_g mode should be weak.

Figure 1. Calculated fatband of dxz , dxy , dyz and dx^2-y^2 orbitals of $Ti_3C_2O_2$ (The diameters of the circles represent the intensity of DOS).

We have added the part “Based on the discussions, the general energy migration pathways are epitomized in Fig. 5b.” in the main text line 321, and “These generated nonthermal electron could interact directly with A_{1g} mode if the τ_{e-ph} is close to or less than the τ_{e-e} , in which the energy of the thermal electron will dissipate to acoustic phonons. If the τ_{e-ph} is much greater than the τ_{e-e} , the energy of nonthermal electron would damp to all kinds of phonons mediated by the thermal electron. In case of a weak screening effect (insufficient screening), the SP decays could couple directly with phonon.” in the caption of Fig.5.

REVIEWER COMMENTS

Reviewer #1 (Remarks to the Author):

I am satisfied by the revision of the authors. I support its publication in Nature Communication in the current form.